# A Dynamic Analysis Method of Liquid-Filled Containers Considering the Fluid–Structure Interaction

Xibing Fang [1,2], Xin Bao [3,*], Fengjiang Yue [4] and Qiyuan Zhao [1,2]

1   Sanya Science and Education Innovation Park, Wuhan University of Technology, Sanya 572024, China; 106953@whut.edu.cn (X.F.); 274714@whut.edu.cn (Q.Z.)
2   School of Civil Engineering and Architecture, Wuhan University of Technology, Wuhan 430070, China
3   State Key Laboratory of Coastal and Offshore Engineering, Dalian University of Technology, Dalian 116000, China
4   Department of Civil Engineering, Xinjiang University, Urumqi 830047, China; yuefjzxj@126.com
*   Correspondence: baoxin@dlut.edu.cn

**Abstract:** Based on acoustic fluid elements, a dynamic analysis of liquid sloshing modes and liquid-filled containers was undertaken, considering the effect of fluid–structure interactions (FSIs). The liquid sloshing modes in two-dimensional (2D) and three-dimensional (3D) containers were analyzed, and the results were compared with liquid sloshing modes measured in tests and theoretically calculated modes. This finding thus verifies the correctness of the simulation method based on acoustic fluid elements. Cylindrical liquid-filled containers with different water levels were subjected to a modal analysis and dynamic and time-history analysis. The results show that the finite element analysis (FEA) based on acoustic fluid elements can accurately simulate liquid sloshing modes in liquid-filled containers, as well as the vibration characteristics of these containers with different liquid levels. The vibration frequency of liquid-filled containers decreases with rising liquid levels. The liquid level significantly affects the distribution of the maximum displacement, maximum acceleration, and maximum von Mises stress on the sidewall of liquid-filled containers. Numerical simulations based on acoustic fluid elements provide an effective and reliable method for dynamic analysis of liquid-filled containers considering the effect of FSIs.

**Keywords:** fluid–structure interaction; acoustic fluid elements; liquid sloshing modes; liquid-filled containers; dynamic analysis

## 1. Introduction

The effect of fluid–structure interactions (FSIs) between solid structures and liquids is prevalent in engineering practice. For example, liquid-filled containers under seismic action or other vibration loads are a structure with the typical effect of FSIs [1]. Analysis of the FSI effect of liquid-filled containers, on the one hand, focuses on liquid sloshing therein, and on the other, considers the influence of liquid sloshing on these containers.

In order to tackle liquid sloshing problems, Dodge et al. [2–5] systematically expounded the theoretical and engineering applications of liquid sloshing modes. However, these methods pay more attention to theoretical and analytical methods and are only applicable to solving cases with regular shapes and simple external excitations, yet fail to solve vibration problems of liquid-filled structures of complex shapes [6,7]. Bao et al. [8,9] studied liquid modes based on potential-based fluid elements. When using this method, the liquid sloshing frequency is much lower than the structural vibration frequency, and the resulting first hundreds of modes are all liquid sloshing modes rather than modes of solid structures [10]. Therefore, it is difficult to apply the mode-superposition response spectrum method and time history analysis to any dynamic analysis of liquid-filled structures, and these methods are inapplicable to the analysis of dynamic responses of liquid-filled containers.

Structural engineers generally pay more attention to how liquid motion affects structural responses under seismic action, while they are not interested in the motion of liquids themselves. To consider the influence of liquid sloshing on the dynamic responses of liquid-filled containers and avoid complex liquid sloshing computation, the simplified equivalent mechanical model of fluid sloshing is generally used. On the basis of the potential flow theory, Graham [11] took the lead to establish an equivalent mechanical model of fluid sloshing in a rectangular container, which has been widely applied in the engineering field. Housner [12] deduced a simplified calculation formula based on an equivalent model of fluids based on physical intuition. The Housner model is a good approximation for the exact solution to the model proposed by Graham and has found extensive applications in civil and hydraulic engineering; however, the Housner model is not based on physical intuition and therefore is not an exact physical model, so its results are unreliable in some cases. Li [13] improved the Housner model on the basis of the theory of linear potential flow and provided a fitting solution to the equivalent model using a semi-analytical and semi-numerical method, whereas the improved Housner model has such complex parameters that it is only suitable for regular two-dimensional (2D) models and not complex 2D and three-dimensional (3D) models. Apart from proposing the equivalent methods based on the potential flow theory, Wstergaard [14] and Chopra [15] also developed the added-mass method. Rajasankar et al. [16] applied the added-mass method to the finite element method (FEM) and performed FSI analysis. Bao et al. [17] proposed an improved added-mass model based on the added-mass method and conducted dynamic analysis on an annular tank, providing reference for the design and application thereof. However, the distributed mass coefficient is difficult to determine in the above added-mass method and improved added-mass methods, so when using these methods to analyze structures of liquid-filled containers, the results are generally less reliable.

The conventional analytical methods are only applicable to cases with regular geometric shapes and simple external excitations, and FEM based on potential-based fluid elements is not applicable to the analysis of the dynamic responses of liquid-filled structures. Moreover, equivalent mechanical models based on fluid sloshing and the added-mass methods also have drawbacks. Considering this, it is necessary to use a method that is not only suitable for exploring liquid sloshing modes but is also applicable to assessing influences of liquid sloshing on the dynamic responses of liquid-filled containers, thus providing a reference for the engineering design and application of such containers. Therefore, the current research conducted a finite element analysis (FEA) on liquid sloshing modes at first and compared the analysis results with theoretical solutions and liquid sloshing frequencies and modes measured in previous tests, thus verifying the correctness of the FEM. However, the theoretical solutions and test models are only applicable to 2D models while the practical models are all 3D ones, so 3D models of liquid sloshing were also analyzed. Then, a dynamic analysis was conducted on liquid-filled containers with different liquid levels, considering the influences of the effect of FSIs and the intrinsic frequency and dynamic response of liquid-filled containers. The FEM based on acoustic fluid elements used in this research provides an effective and reliable method for the dynamic analysis of liquid-filled containers considering the effect of FSIs.

## 2. The FEA Theory Based on Acoustic Fluid Elements

A liquid-filled container and a liquid constitute an FSI system, in which the liquid and structure (liquid-filled container) are simulated using FEM. The FEM simulation methods of structures have been introduced elsewhere [18]. For FEA considering the effect of FSIs, one should use the fluid displacement as an unknown quantity [19] and harness the similarity between the fluid motion equation and the equation of motion of structural elastomers, which results in a finite element calculation model of fluid consistent with the finite element scheme; the other is to take the fluid pressure as an unknown quantity [20] and coordinate the displacement and pressure on the structure–fluid interface, from which the obtained mass and stiffness matrices are asymmetric matrices. When analyzing liquid sloshing

problems using the pressure scheme based on acoustic fluid elements [21], the theory is as described below.

For the structure,

$$[M_S]\{\ddot{u}\} + [C_S]\{\dot{u}\} + [K_S]\{u\} - [R]\{p\} = \{f_S\} \tag{1}$$

where $[M_S], [C_S], [K_S]$ separately represent the mass, damping, and stiffness matrices of the structure; $\{\ddot{u}\}, \{\dot{u}\}, \{u\}$ separately denote the acceleration, speed, and displacement vectors at nodes of structural elements; $[R]$ denotes the coupling matrix at the structure–fluid interface; $\{p\}$ is the nodal pressure vector of fluid elements; and $\{f_S\}$ is the load vector of the structure.

For the fluid,

$$[M_F]\{\ddot{p}\} + [C_F]\{\dot{p}\} + [K_F]\{p\} + \bar{\rho}_0[R]^T\{\ddot{u}\} = \{f_F\} \tag{2}$$

where $[M_F], [C_F],$ and $[K_F]$ separately represent the mass, damping, and stiffness matrices of the fluid; $\{\dot{p}\}, \{\ddot{p}\}$ are the first-order and second-order derivatives of nodal pressure of fluid elements; $\bar{\rho}_0$ denotes the fluid density; and $\{f_F\}$ is the load vector of the fluid.

The following can be obtained by combining Equations (1) and (2):

$$\begin{bmatrix} [M_S] & 0 \\ \bar{\rho}_0[R]^T & [M_F] \end{bmatrix} \begin{Bmatrix} \{\ddot{u}\} \\ \{\ddot{p}\} \end{Bmatrix} + \begin{bmatrix} [C_S] & 0 \\ 0 & [C_F] \end{bmatrix} \begin{Bmatrix} \{\dot{u}\} \\ \{\dot{p}\} \end{Bmatrix} + \begin{bmatrix} [K_S] & -[R] \\ 0 & [K_F] \end{bmatrix} \begin{Bmatrix} \{u\} \\ \{p\} \end{Bmatrix} = \begin{Bmatrix} \{f_S\} \\ \{f_F\} \end{Bmatrix} \tag{3}$$

By solving Equation (3), the liquid sloshing modes in the FSI system and the FSI dynamic responses can be obtained. The bounding surface of the water domain are modeled using various types of interface defined as follows:

(i) *Fluid–structure interaction:* Set a fluid–structure interaction on the boundary between fluid and structure.

(ii) *Free surface:* Set a free surface on the top surface of water domain, which can provide an approximate representation of the water surface wave.

(iii) *Rigid wall:* Set a rigid wall on the bottom and the lateral surface where the water cannot flow through the boundary.

As explained above, the surface of the water domain is a defined, specified boundary condition. In general, the software ANSYS 2023 can automatically generate fluid–structure interactions along the boundary between the fluid and structure.

## 3. Analysis of Liquid Sloshing Modes

### 3.1. Theoretical Solutions to 2D Liquid Sloshing Modes

Following the potential flow theory of ideal fluids, the fluid is assumed to be a non-viscous, irrotational, and incompressible ideal fluid, for which the influence of the free surface tension of the liquid is not considered. For general practical engineering problems, the above assumption is feasible. The corresponding eigenvalue equation is established by establishing the free sloshing equation for the liquid using the theoretical method. The liquid sloshing frequency and mode are attained by finding the eigenvalues.

3.1.1. Two-Dimensional Rectangular Container

For a regular rectangular container (Figure 1a), the *j*th sloshing frequency of the liquid is [2,3]:

$$\omega_j = \sqrt{g\left(\frac{j\pi}{2a}\right)\tanh\left(\frac{j\pi h}{2a}\right)} \quad j = 1, 2, 3, \ldots \tag{4}$$

where g is the gravitational acceleration; *a* represents the half width of the rectangular container; and *h* is the liquid level. According to Equation (4), the sloshing of the free liquid surface is attributed to gravity. The *j*th sloshing mode of the free liquid surface is

$$\phi_j(x,\, z) = \begin{cases} C_j \sin \frac{j\pi}{2a} x & j = 1,\, 3,\, 5,\, \dots \text{ (antisymmetric mode)} \\ D_j \cos \frac{j\pi}{2a} x & j = 2,\, 4,\, 6,\, \dots \text{ (symmetric mode)} \end{cases} \tag{5}$$

where $C_j$ and $D_j$ are constants. Figure 2 illustrates the first six sloshing modes of the free liquid surface. Therein, odd-order frequencies $\omega_1$, $\omega_3$, $\omega_5$ correspond to the first three antisymmetric modes, while even-order frequencies $\omega_2$, $\omega_4$, $\omega_6$ correspond to the first three symmetric modes, respectively.

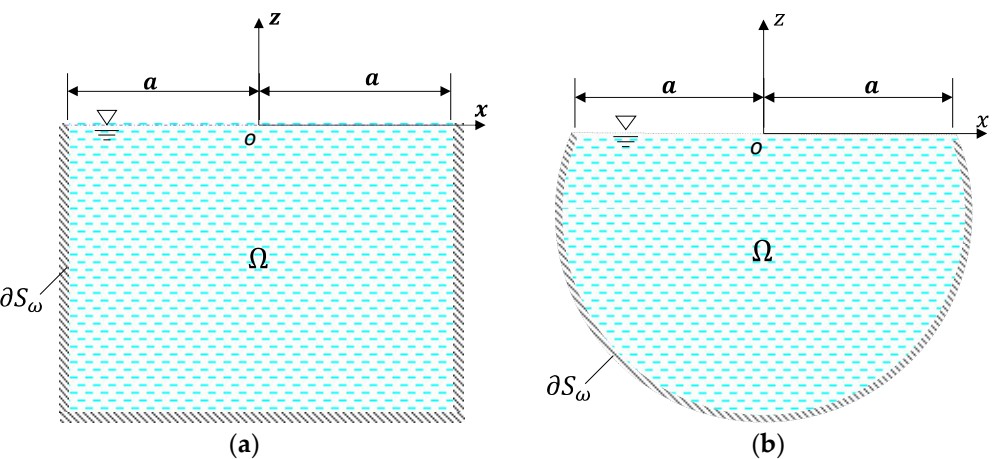

**Figure 1.** Two-dimensional containers. (**a**) Rectangular container. (**b**) Arbitrary section container.

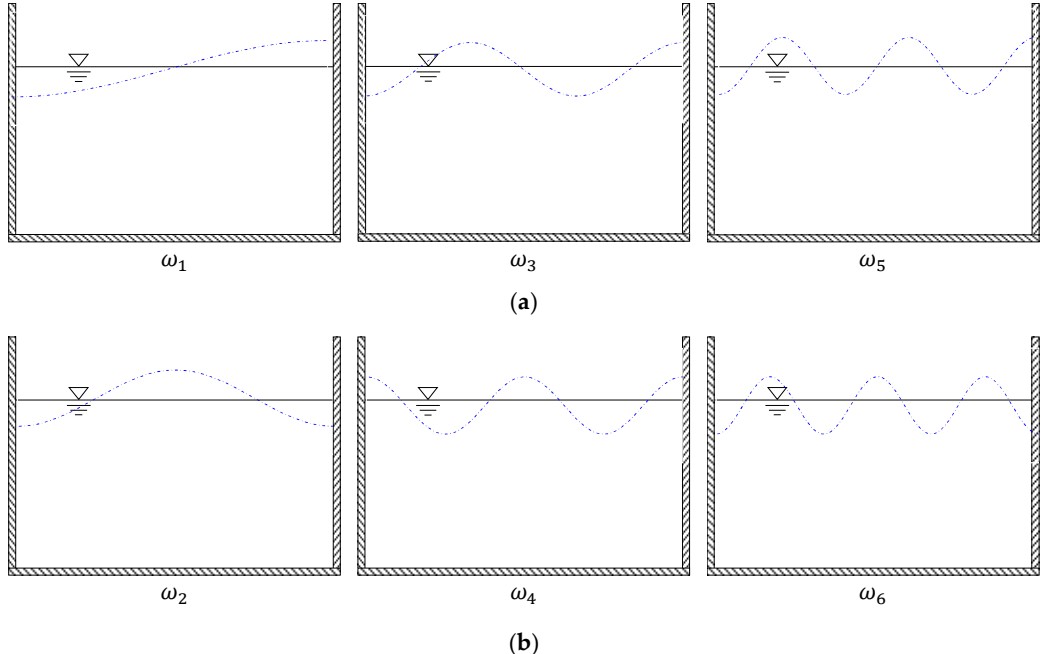

**Figure 2.** The first six sloshing modes of the free liquid surface in the rectangular container. (**a**) Antisymmetric (odd-order) modes ($j = 1, 3, 5$). (**b**) Symmetric (even-order) modes ($j = 2, 4, 6$).

### 3.1.2. Container with Arbitrary Cross Sections

A container of arbitrary shape is displayed in Figure 1b. It is challenging to solve the theoretical solution due to the irregular boundary. Considering this, the Ritz method is adopted to calculate the liquid sloshing frequency of containers of arbitrary shape [22]:

$$\omega_j^2 \cong \begin{cases} B_j \alpha_j g & j = 1,\ 3,\ 5,\ \ldots \quad (\text{antisymmetric}) \\ D_j \alpha_j g & j = 2,\ 4,\ 6,\ \ldots \quad (\text{symmetric}) \end{cases} \tag{6}$$

$$B_j = \frac{-b_{Aj} - \sqrt{b_{Aj}^2 - 4a_{Aj}c_{Aj}}}{2a_{Aj}} \quad j = 1,3,5,\ldots \tag{7}$$

$$D_j = \frac{-b_{Sj} - \sqrt{b_{Sj}^2 - 4a_{Sj}c_{Sj}}}{2a_{Sj}} \quad j = 2,4,6,\ldots \tag{8}$$

$$\begin{cases} a_{Aj} = \iint_\Omega \left( \sin^2 \alpha_j x + \sinh^2 \alpha_j z \right) dx dz \\ b_{Aj} = 2 \iint_\Omega \left( \sinh \alpha_j z \cdot \cosh \alpha_j z \right) dx dz - a/\alpha_j \quad j = 1,3,5,\ldots \\ c_{Aj} = \iint_\Omega \left( \cos^2 \alpha_j x + \sinh^2 \alpha_j z \right) dx dz \end{cases} \tag{9}$$

$$\begin{cases} a_{Sj} = \iint_\Omega \left( \cos^2 \alpha_j x + \sinh^2 \alpha_j z \right) dx dz \\ b_{Sj} = 2 \iint_\Omega \left( \sinh \alpha_j z \cdot \cosh \alpha_j z \right) dx dz - a/\alpha_j \quad j = 2,4,6,\ldots \\ c_{Sj} = \iint_\Omega \left( \sin^2 \alpha_j x + \sinh^2 \alpha_j z \right) dx dz \end{cases} \tag{10}$$

where $g$ denotes the gravitational acceleration; $\alpha_j = j\pi/2a$; $a$ is the half width of the resting free liquid surface; and $\Omega$ is the liquid area.

### 3.2. FEA of 2D Liquid Sloshing Modes

The model adopted is a flat container, through which the 2D liquid sloshing is simulated. The test models and the finite element models are illustrated in Figures 3 and 4, respectively. The four-node acoustic fluid elements are used to model the water domain. The mesh densities of all finite element models were refined until the convergence of the results. Among them, the mesh size of the rectangular container is 0.005 m, and the mesh size of the circular container and U-shaped container along the diameter direction is 0.005 m. The water level and half width of the rectangular container are $h$ = 0.12 m and $a$ = 0.10 m; the water level and radius of the circular container are $h$ = 0.16 m and $R$ = 0.125 m; and the radius, water level, and cavity thickness of the U-shaped container are $R$ = 0.10 m, $h$ = 0.115 m, and 20 mm, respectively [23].

In the FEA, the containers are assumed to be rigid bodies. Generally, containers are all elastomers, while the elastic vibration of containers only slightly influences the overall liquid sloshing, so the influence of the stiffness of containers on liquid sloshing is ignored when analyzing liquid sloshing modes.

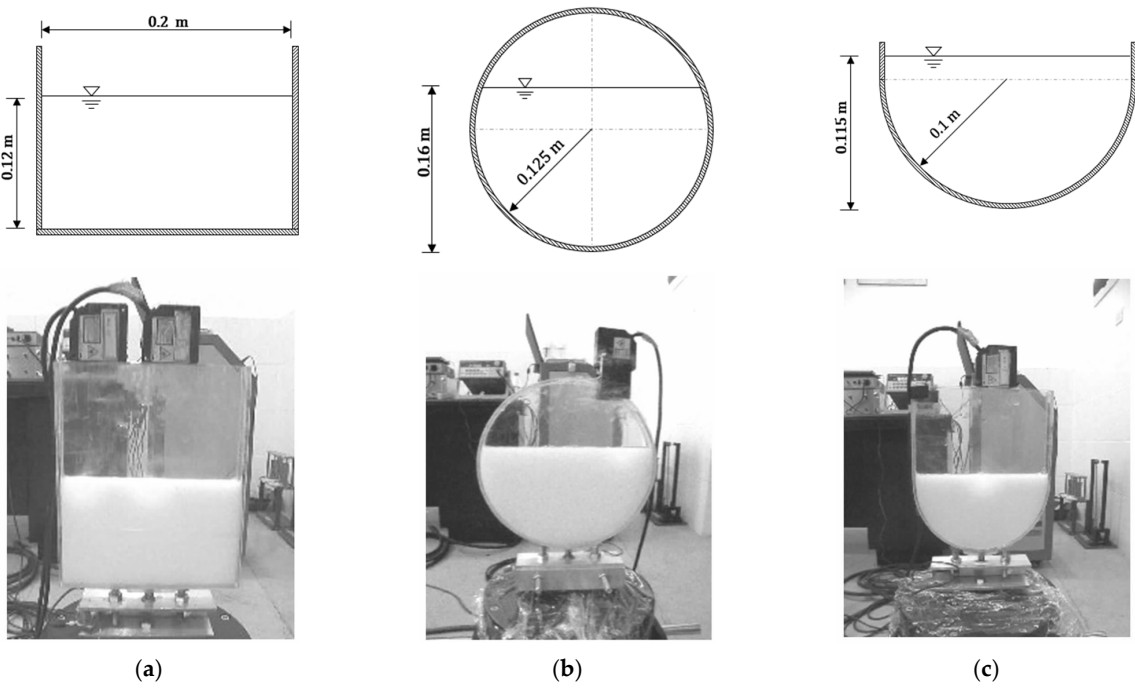

**Figure 3.** Test models [23,24]. (**a**) Rectangular container. (**b**) Circular container. (**c**) U-shaped container.

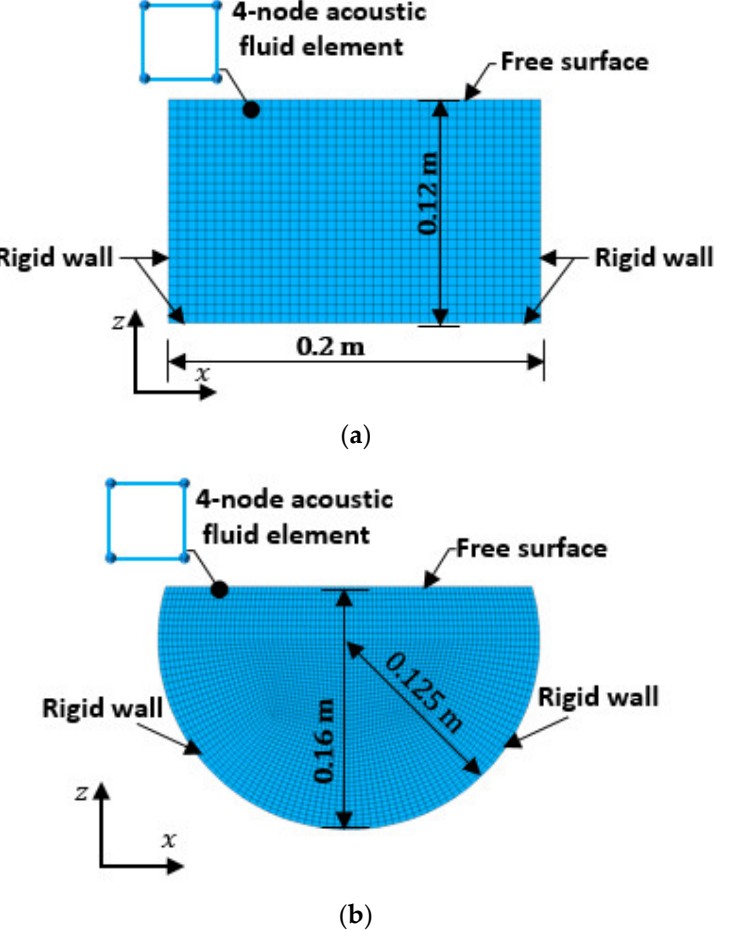

**Figure 4.** *Cont.*

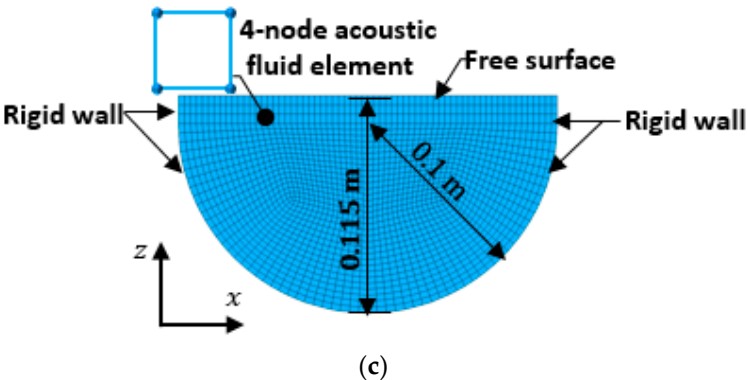

**(c)**

**Figure 4.** Finite element models. (**a**) Rectangular container. (**b**) Circular container. (**c**) U-shaped container.

### 3.2.1. Rectangular Container

The first four sloshing modes in the rectangular container obtained by FEA are shown in Figure 5; those measured in tests are illustrated in Figure 6. Table 1 presents that the liquid sloshing frequency attained by FEA is consistent with the results of theoretical calculation, with the maximum error within 0.5%. The maximum error in the sloshing frequency attained by FEA and that measured in tests is 3.1%. One of the causes for the error in the liquid sloshing frequency obtained using FEA and tests is that the liquid is assumed to be an ideal fluid in FEA, that is, incompressible fluid without viscosity, while the liquid in the tests is viscous. Furthermore, there is also an error (albeit small) arising in the test process. A comparison of Figure 2 with Figures 5 and 6 shows that the sloshing modes obtained by FEA agree well with the theoretically calculated modes, with a small difference in only the fourth order. The fourth sloshing modes obtained by the FEA and experiment are both even-order symmetric modes (Figure 2), so they are not, in the theoretical sense, different.

**Table 1.** Comparison of the first four orders in FEA, theoretical analysis, and tests (unit: Hz).

| Order | Rectangular Container Water Level $h$ = 0.12 m | | | Circular Container Water Level $h$ = 0.16 m | | | U-Shaped Container Water Level $h$ = 0.115 m | | |
|---|---|---|---|---|---|---|---|---|---|
| | Theoretical Calculation | Test | FEA | Theoretical Calculation | Test | FEA | Theoretical Calculation | Test | FEA |
| 1 | 1.93 | 1.89 | 1.93 | 1.77 | 1.68 | 1.77 | 1.90 | 1.85 | 1.89 |
| 2 | 2.79 | 2.73 | 2.79 | 2.55 | 2.52 | 2.57 | 2.78 | 2.73 | 2.78 |
| 3 | 3.42 | 3.40 | 3.43 | 3.12 | 3.09 | 3.15 | 3.42 | 3.35 | 3.42 |
| 4 | 3.95 | 3.94 | 3.97 | 3.61 | 3.51 | 3.64 | 3.95 | 3.87 | 3.95 |

Note: Test data in the table are taken from [23,24].

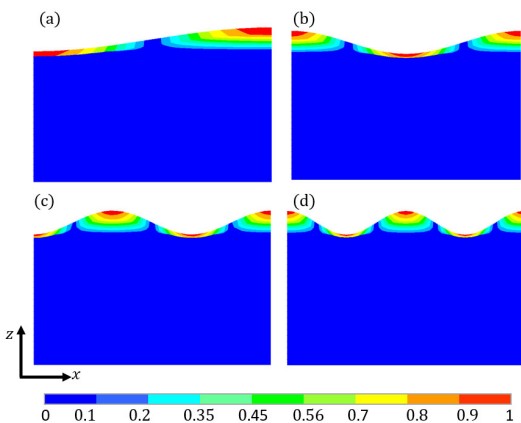

**Figure 5.** The first four liquid sloshing modes by FEA: (**a**) 1st; (**b**) 3rd; (**c**) 6th; (**d**) 9th.

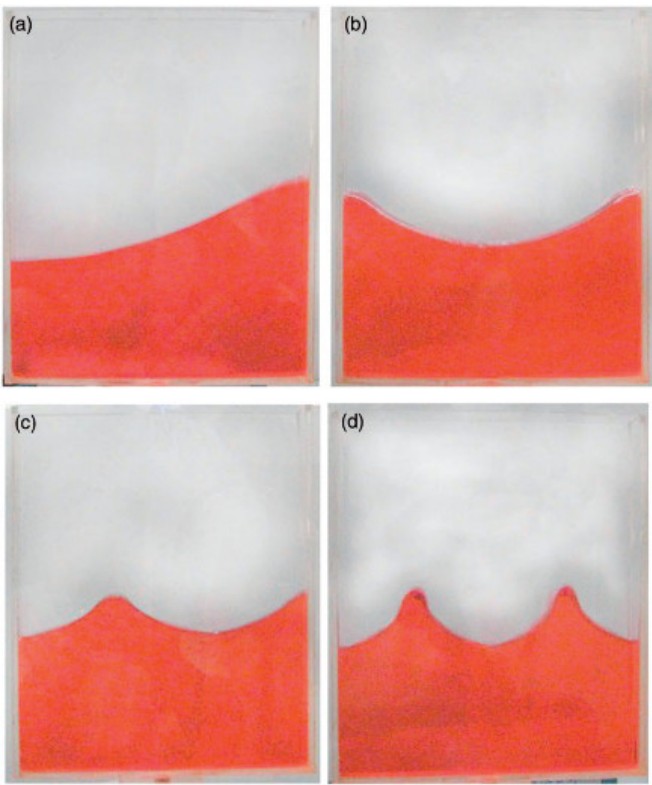

**Figure 6.** The first four liquid sloshing modes in the tests [23,24]: (**a**) 1st; (**b**) 2nd; (**c**) 3rd; (**d**) 4th.

### 3.2.2. Circular Container

The first four liquid sloshing modes in the circular container obtained by FEA are illustrated in Figure 7; those obtained in the tests are shown in Figure 8. Their comparison shows that the sloshing modes obtained by FEA match those in the tests, with a small difference in only the second order, whereas the second modes are both symmetric modes, so they can be regarded, in the theoretical sense, as consistent. Table 1 shows that the sloshing frequencies obtained by FEA are basically consistent with the first four sloshing frequencies attained by theoretical calculation, with the maximum error of 1.4%. The sloshing frequencies acquired by FEA have the maximum discrepancy of 5% with those measured experimentally, the source of which can be found in the error analysis as applied to the rectangular container.

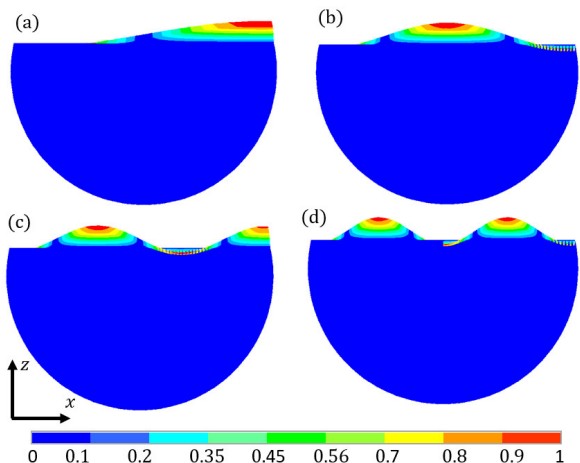

**Figure 7.** The first four liquid sloshing modes by FEA: (**a**) 1st; (**b**) 2nd; (**c**) 3rd; (**d**) 4th.

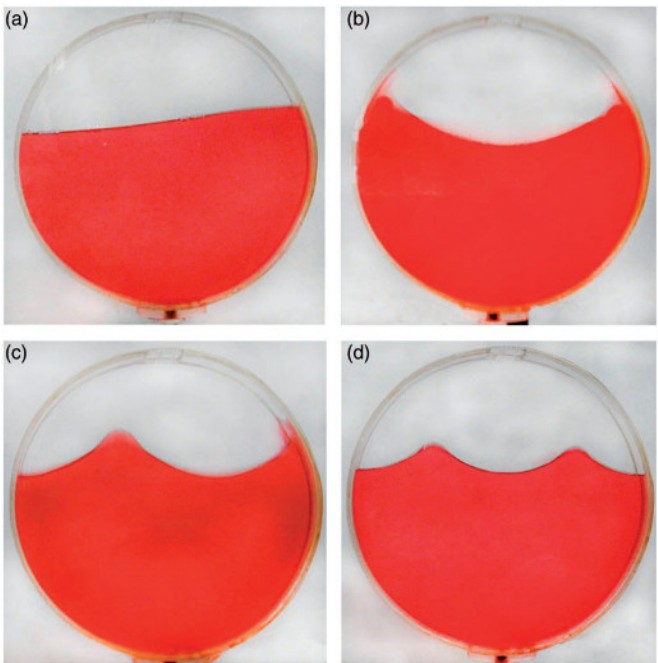

**Figure 8.** The first four liquid sloshing modes in the tests [23,24]: (**a**) 1st; (**b**) 2nd; (**c**) 3rd; (**d**) 4th.

### 3.2.3. U-Shaped 2D Container

The first four sloshing modes in the U-shaped 2D container attained by FEA are illustrated in Figure 9, while those measured in the tests are shown in Figure 10. Their comparison shows that the sloshing modes attained by FEA are consistent with those measured in the tests, with a subtle difference in only the fourth order, whereas the fourth modes are both symmetric modes, which are regarded as similar (in the theoretical sense). As displayed in Table 1, the sloshing frequencies predicted by FEA match those theoretically calculated for the first four frequencies, with differences of less than 0.5%. The sloshing frequencies obtained by FEA and those measured experimentally differ by no more than 2%, which indicates that the FEA results are reliable (again, the error analysis follows that applied to the rectangular container).

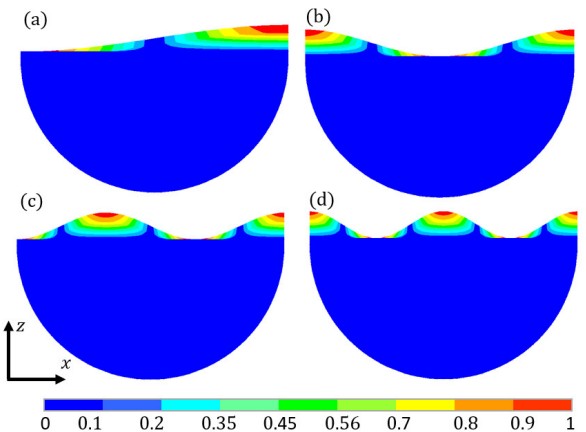

**Figure 9.** The first four liquid sloshing modes by FEA: (**a**) 1st; (**b**) 2nd; (**c**) 3rd; (**d**) 4th.

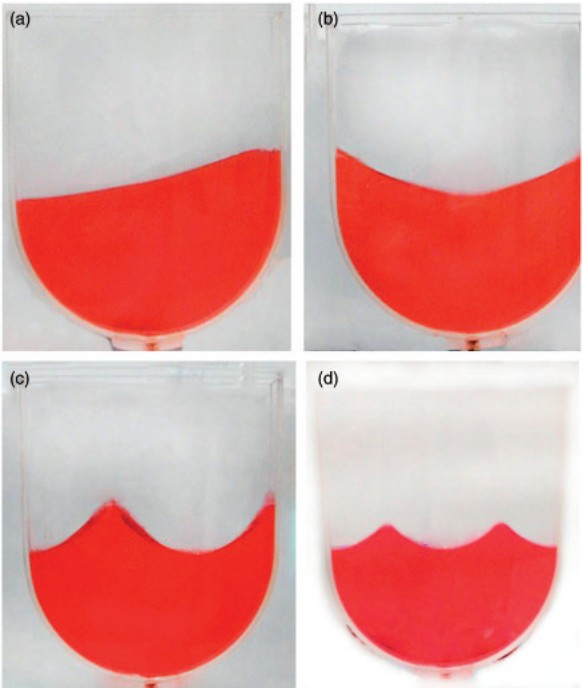

**Figure 10.** The first four liquid sloshing modes in the tests [23,24]: (**a**) 1st; (**b**) 2nd; (**c**) 3rd; (**d**) 4th.

Based on the above analysis results, the liquid sloshing frequencies conform well to the theoretical calculation and test results, despite small differences in several modes. This outcome indicates that the FEM based on acoustic fluid elements can be used to simulate the liquid sloshing modes in containers of different shapes.

### 3.3. FEA of 3D Liquid Sloshing Modes

To compare with the 2D models, the models of 3D containers were assigned identical geometrical dimensions to their 2D counterparts. The 3D diagram models and the 3D finite element models are shown in Figures 11 and 12, respectively. Compared to the 2D finite element model, the eight-node acoustic fluid elements are used to model the water domain, and the element size is reduced for enhancing computational efficiency. Among them, the mesh size of the rectangular container is 0.01 m, and the mesh size of the circular container and U-shaped-container along the diameter direction is 0.01 m. For the cuboid container, the water level is $h = 0.12$ m, and its length and width are $a = b = 0.20$ m; the water level and radius of the spherical container are $h = 0.16$ m and $R = 0.125$ m; the water level and radius at the bottom of the U-shaped 3D container are $h = 0.115$ m and $R = 0.10$ m, respectively.

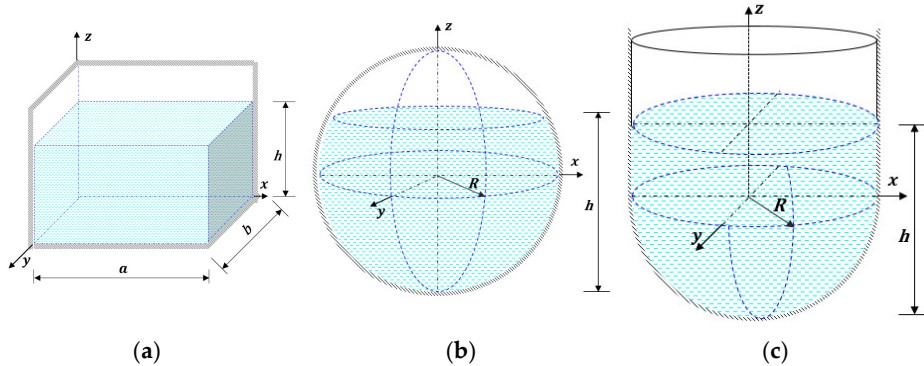

(**a**)    (**b**)    (**c**)

**Figure 11.** Three-dimensional diagram models. (**a**) Cuboid container. (**b**) Spherical container. (**c**) U-shaped 3D container.

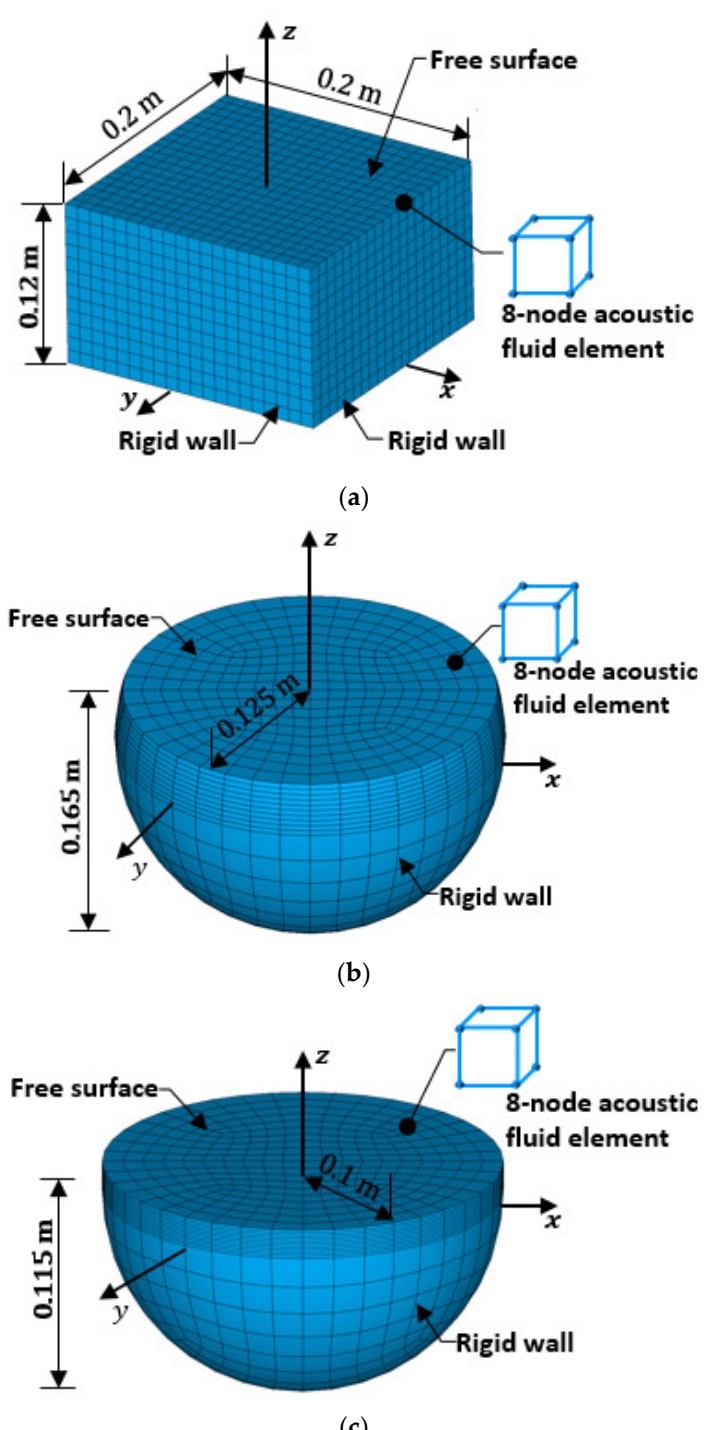

**Figure 12.** Three-dimensional finite element models. (**a**) Cuboid container. (**b**) Spherical container. (**c**) U-shaped 3D container.

### 3.3.1. Cuboid Container

The liquid sloshing modes in the cuboid container obtained by FEA are shown in Figures 13 and 14. Comparisons of Figures 5 and 13, combined with Tables 1 and 2, show that the first, third, sixth, and ninth liquid sloshing modes in the cuboid container separately correspond to the first, second, second, and fourth liquid sloshing modes in the rectangular container. The difference is that the liquid sloshing modes in the cuboid container are more complex than those in the rectangular container, and the frequencies are higher. It is more difficult to resolve the liquid sloshing modes in the cuboid container than those in the

rectangular container in the theoretical sense. Therefore, the FEM based on acoustic fluid elements can better simulate liquid sloshing modes in the cuboidal container.

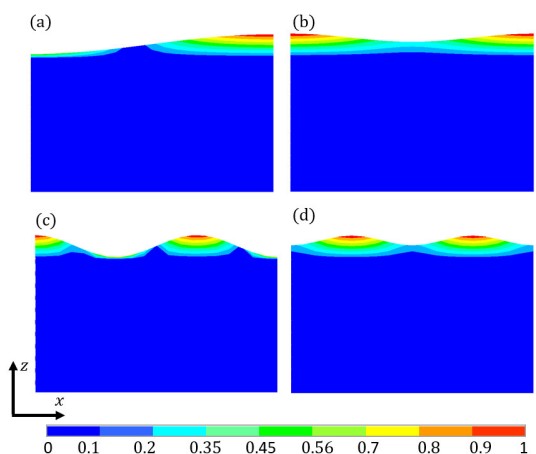

**Figure 13.** Two-dimensional view of liquid sloshing modes: (**a**) 1st; (**b**) 3rd; (**c**) 6th; (**d**) 9th.

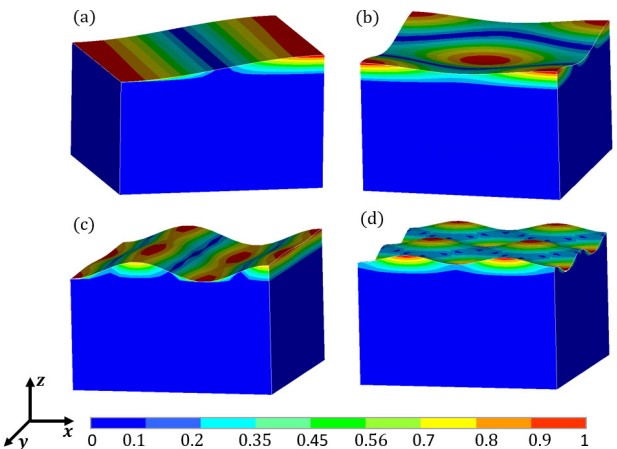

**Figure 14.** Three-dimensional view of liquid sloshing modes: (**a**) 1st; (**b**) 3rd; (**c**) 6th; (**d**) 9th.

**Table 2.** The first ten liquid sloshing frequencies (unit: Hz).

| Container | Order | | | | | | | | | |
|---|---|---|---|---|---|---|---|---|---|---|
| | 1 | 2 | 3 | 4 | 5 | 6 | 7 | 8 | 9 | 10 |
| Cuboid container (Water level $h$ = 0.12 m) | 1.93 | 2.34 | 2.80 | 2.97 | 3.34 | 3.45 | 3.55 | 3.79 | 4.01 | 4.08 |
| Spherical container (Water level $h$ = 0.16 m) | 1.95 | 2.57 | 2.85 | 3.04 | 3.37 | 3.44 | 3.79 | 3.80 | 3.88 | 4.12 |
| U-shaped 3D container (Water level $h$ = 0.115 m) | 2.04 | 2.72 | 3.09 | 3.27 | 3.65 | 3.66 | 4.04 | 4.11 | 4.22 | 4.40 |

### 3.3.2. Spherical Container

The liquid sloshing modes in the spherical container acquired by FEA are shown in Figures 15 and 16. Comparisons of Figures 7 and 15, combined with Tables 1 and 2, show that, different from the cuboid and rectangular containers, the liquid sloshing modes in the spherical container do not correspond to those in the spherical container. This is because the length of the free liquid surface in the cuboid container ($a = b$ = 0.20 m) remains identical to that in the rectangular container ($a$ = 0.20 m), while the free liquid surface in the spherical container is not geometrically comparable to that in the spherical container.

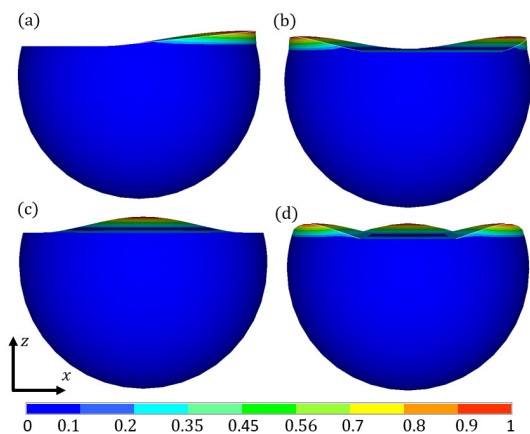

**Figure 15.** Two-dimensional view of liquid sloshing modes: (**a**) 1st; (**b**) 2rd; (**c**) 3rd; (**d**) 4th.

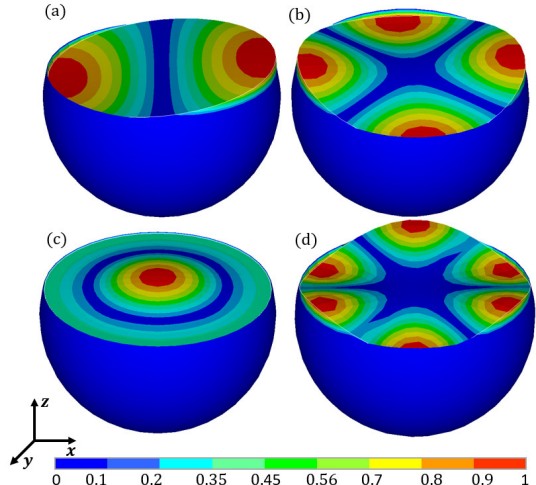

**Figure 16.** Three-dimensional view of liquid sloshing modes: (**a**) 1st; (**b**) 2rd; (**c**) 3rd; (**d**) 4th.

### 3.3.3. U-Shaped 3D Container

The liquid sloshing modes in the U-shaped 3D container obtained by FEA are shown in Figures 17 and 18. By comparing Figures 9 and 17 and combining with Tables 1 and 2, the liquid sloshing modes in the U-shaped 3D container do not correspond to those in the U-shaped 2D container, while they are similar to those in the spherical container. This is because the free liquid surface in the U-shaped 3D container demonstrates geometric similarity to that in the spherical container.

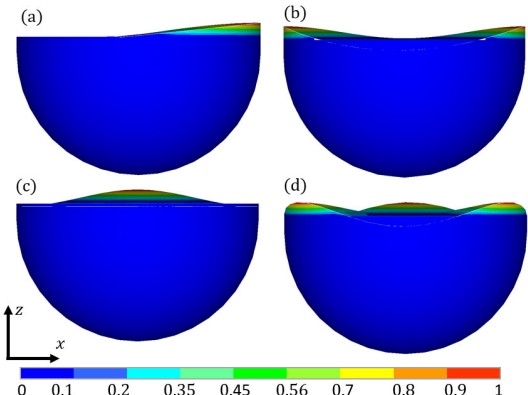

**Figure 17.** Two-dimensional view of liquid sloshing modes: (**a**) 1st; (**b**) 2rd; (**c**) 3rd; (**d**) 4th.

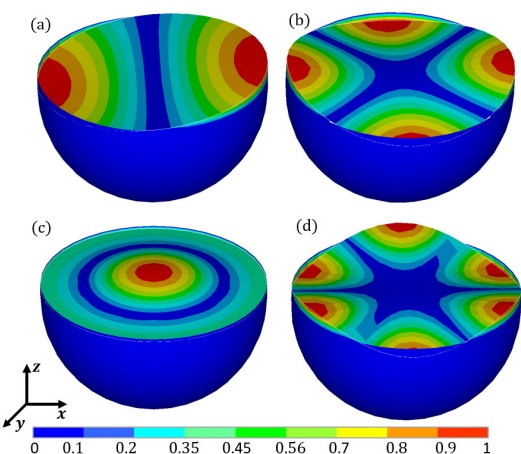

**Figure 18.** Three-dimensional view of liquid sloshing modes: (**a**) 1st; (**b**) 2rd; (**c**) 3rd; (**d**) 4th.

A comparison of the liquid sloshing modes in the 2D containers with those in the 3D containers reveals that the liquid sloshing modes in the 3D containers are more complex than those in the 2D containers, and the modes do not correspond to each other. The FEM based on acoustic fluid elements can be used to simulate liquid sloshing modes in 2D and 3D containers of different shapes.

## 4. Modal Analysis and Time-Historical Analysis of Cylindrical Liquid-Filled Containers

### 4.1. Liquid Modal Analysis

For the cylindrical liquid-filled container in Figure 19, the liquid sloshing modes in elastic and rigid containers were analyzed separately. As illustrated in Figure 20, the eight-node acoustic fluid elements are used to model the water domain, and the eight-node solid elements are used to model the cylindrical container shell. The model sizes and material parameters are listed in Table 3 [8].

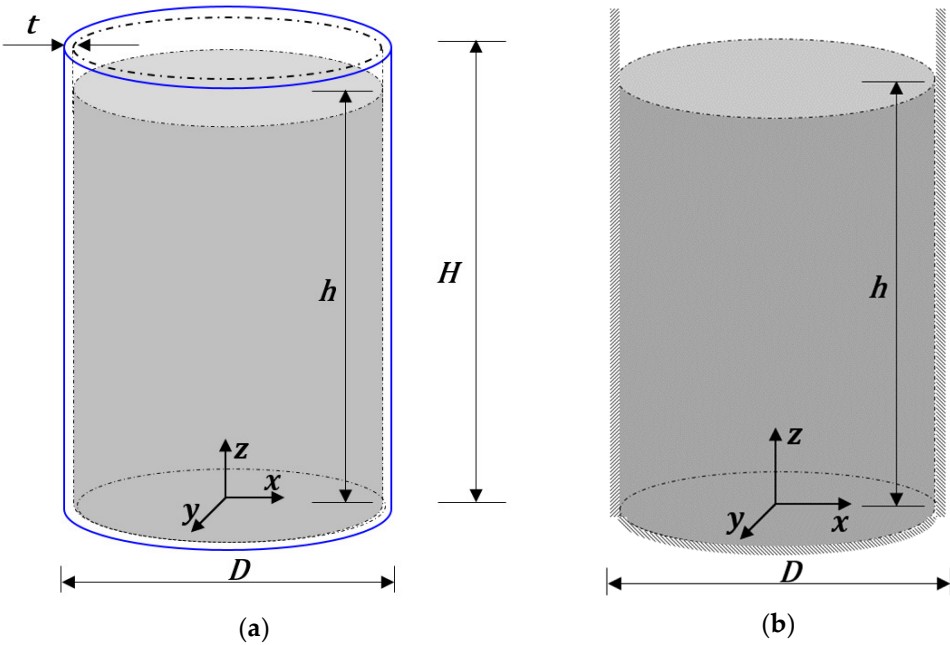

**Figure 19.** Cylindrical liquid-filled containers. (**a**) Elastic container. (**b**) Rigid container.

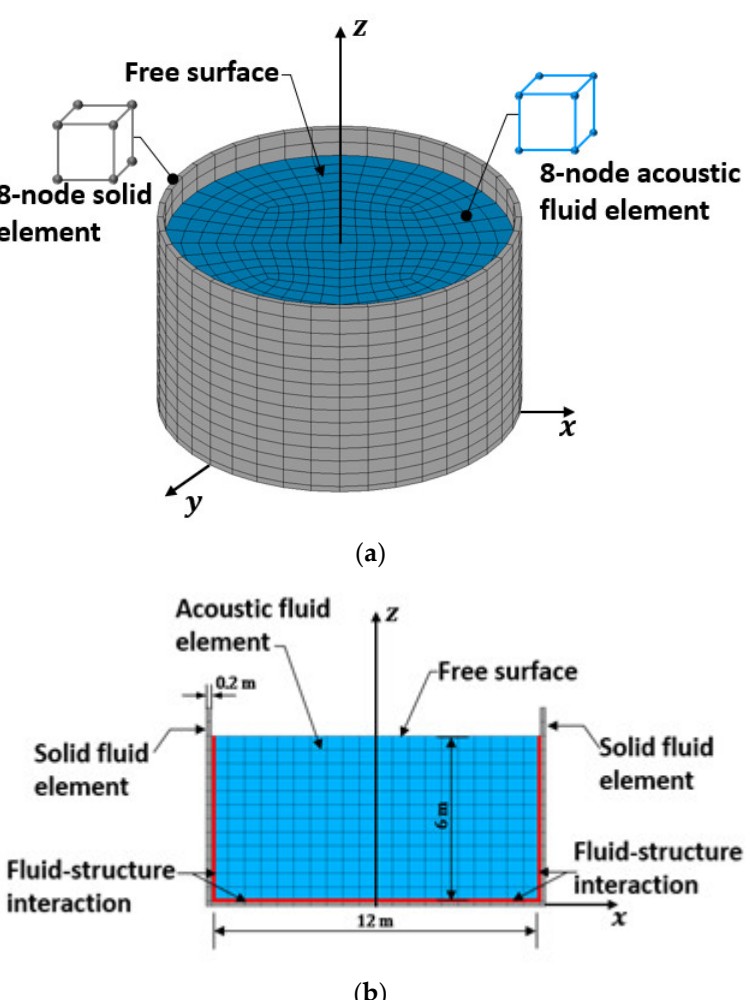

**Figure 20.** Finite element model of elastic container: (**a**) 3D view. (**b**) Two-dimensional elevation through a central cutting plane.

**Table 3.** Model sizes and material parameters.

| Physical Qualities | Symbol | Value |
|---|---|---|
| Container diameter | $D$ | 12 m |
| Container height | $H$ | 7 m |
| Container wall thickness | $t$ | 0.2 m |
| Liquid level | $h$ | 6 m |
| Container elastic modulus | $E$ | $3.10 \times 10^{10}$ Pa |
| Container density | $\rho_1$ | 2643 kg m$^{-3}$ |
| Container Poisson's ratio | $\mu$ | 0.15 |
| Liquid density | $\rho_2$ | 1000 kg m$^{-3}$ |
| Liquid acoustic velocity | $c$ | 1435 m s$^{-1}$ |

The liquid sloshing frequency in the cylindrical liquid-filled container is [2,3]

$$\omega_i = \sqrt{\lambda_i \frac{g}{R} \tanh\left(\lambda_i \frac{h}{R}\right)} \tag{11}$$

where $\lambda_i$ is the *i*th root of the derivative of the family of Bessel functions; g is the gravitational acceleration; $R$ is the radius of the liquid-filled container; and $h$ denotes the liquid level in the liquid-filled container.

The first five liquid sloshing frequencies in the cylindrical liquid-filled container are listed in Table 4. The liquid sloshing modes in the cylindrical liquid-filled container and the vibration modes of this container are shown in Figures 21 and 22, respectively. The FEA numerical solution conforms to the theoretical solution, which indicates that it is reliable and can be used to simulate liquid sloshing modes in containers. Moreover, the FEA predictions of liquid sloshing frequencies in rigid and elastic containers are same for this example, suggesting that the influence of structural stiffness on the liquid sloshing modes can be ignored.

**Table 4.** First five liquid sloshing frequencies of cylindrical containers (unit: Hz).

|  | Order | | | | |
|---|---|---|---|---|---|
|  | **1** | **2** | **3** | **4** | **5** |
| FEA numerical solution (rigid container) | 0.27 | 0.35 | 0.40 | 0.42 | 0.47 |
| FEA numerical solution (elastic container) | 0.27 | 0.35 | 0.40 | 0.42 | 0.47 |
| Theoretical solution | 0.27 | 0.35 | 0.40 | 0.42 | 0.47 |

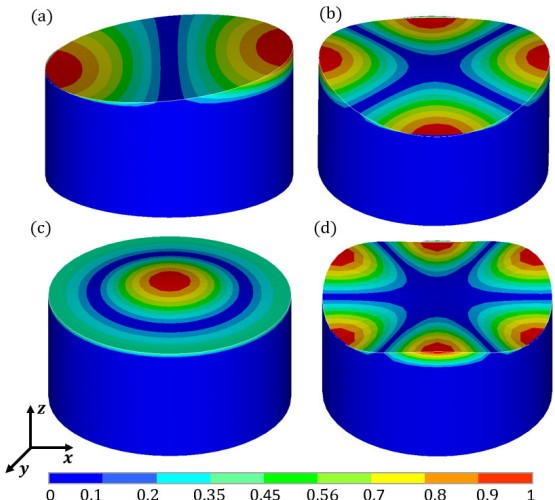

**Figure 21.** Liquid sloshing modes in the cylindrical liquid-filled container: (**a**) 1st; (**b**) 2rd; (**c**) 3rd; (**d**) 4th.

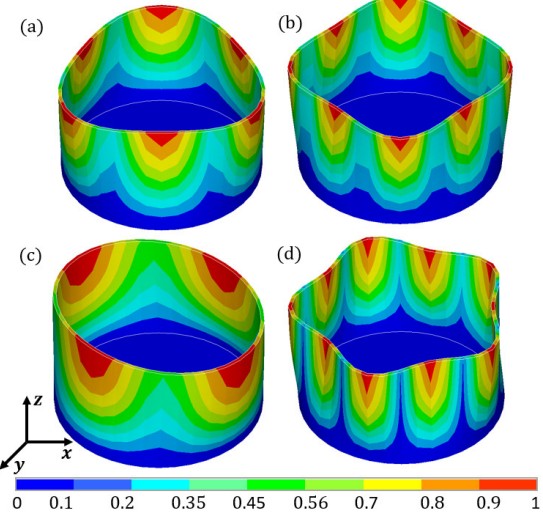

**Figure 22.** Vibration modes of the cylindrical liquid-filled container: (**a**) 1st; (**b**) 2rd; (**c**) 3rd; (**d**) 4th.

### 4.2. Modal Analysis of Cylinder Containers

In practical engineering application, the vibration frequencies of liquid-filled containers themselves attract more attention, in addition to liquid sloshing modes in liquid-filled containers. Therefore, it is necessary to study containers filled to different levels.

In order to measure the influence of different water depths on the frequency of the cylindrical container, the dimensionless coefficient is used to represent the influence coefficient of the frequency of the cylindrical container caused by water depth:

$$R_f = \frac{f_i - f_0}{f_0} \tag{12}$$

where $f_i$ is the frequency of the cylindrical container at different water depths, and $f_0$ is the frequency of the cylindrical container without water.

The first five frequencies of liquid-filled containers with different liquid levels are displayed in Table 5 and Figure 23. The results show that the liquid exerts significant influences on the intrinsic frequency of cylindrical containers. As the liquid level rises in the liquid-filled containers, the vibration frequency of cylindrical containers decreases. If the liquid level in the liquid-filled containers is 6 m, the first frequency reduces by 31.24%. Therefore, the influences of the effect of FSIs on the intrinsic frequency and dynamic response of liquid-filled containers should be considered in the engineering design of such containers.

**Table 5.** The first five frequencies of cylindrical containers (unit: Hz).

| Order | Liquid Level | | | | | | |
|---|---|---|---|---|---|---|---|
| | 0 m | 1 m | 2.0 m | 3.0 m | 4.0 m | 5.0 m | 6.0 m |
| 1 | 19.94 | 19.94 | 19.88 | 19.45 | 18.05 | 15.93 | 13.71 |
| 2 | 20.11 | 20.11 | 20.02 | 19.48 | 18.23 | 16.30 | 14.20 |
| 3 | 26.81 | 26.81 | 26.72 | 26.13 | 24.39 | 21.46 | 18.37 |
| 4 | 29.29 | 29.28 | 29.09 | 27.85 | 24.93 | 21.88 | 19.41 |
| 5 | 38.30 | 38.30 | 38.16 | 37.10 | 34.10 | 30.44 | 26.34 |

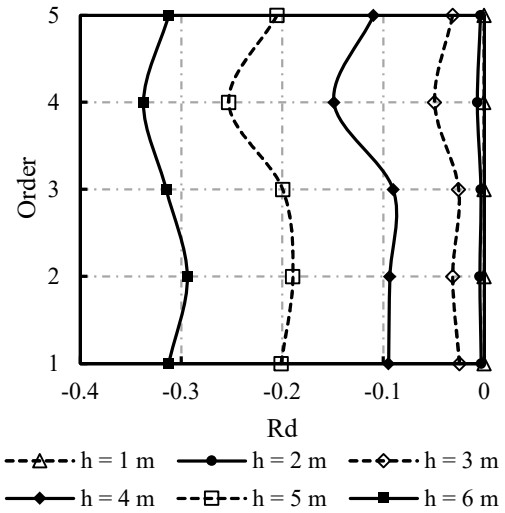

**Figure 23.** The first five $R_f$ of cylindrical containers.

### 4.3. Time-Historical Analysis of Cylindrical Liquid-Filled Containers

Time-historical analysis was conducted by taking cylindrical liquid-filled containers with different liquid levels. The El-Centro (1940), Kobe (1995), and Loma Prieta (1989) waves were selected and applied in the $x$ direction, with the peak acceleration of 0.1 g. The acceleration time-history curves and Fourier spectra are shown in Figure 24. Structural

analysis generally focuses on the displacement, acceleration, and stress of structures. Hence, the maximum displacement, maximum acceleration, and maximum von Mises stress on the sidewall at different heights of liquid-filled containers at $x = D/2$ and $y = 0$ were monitored.

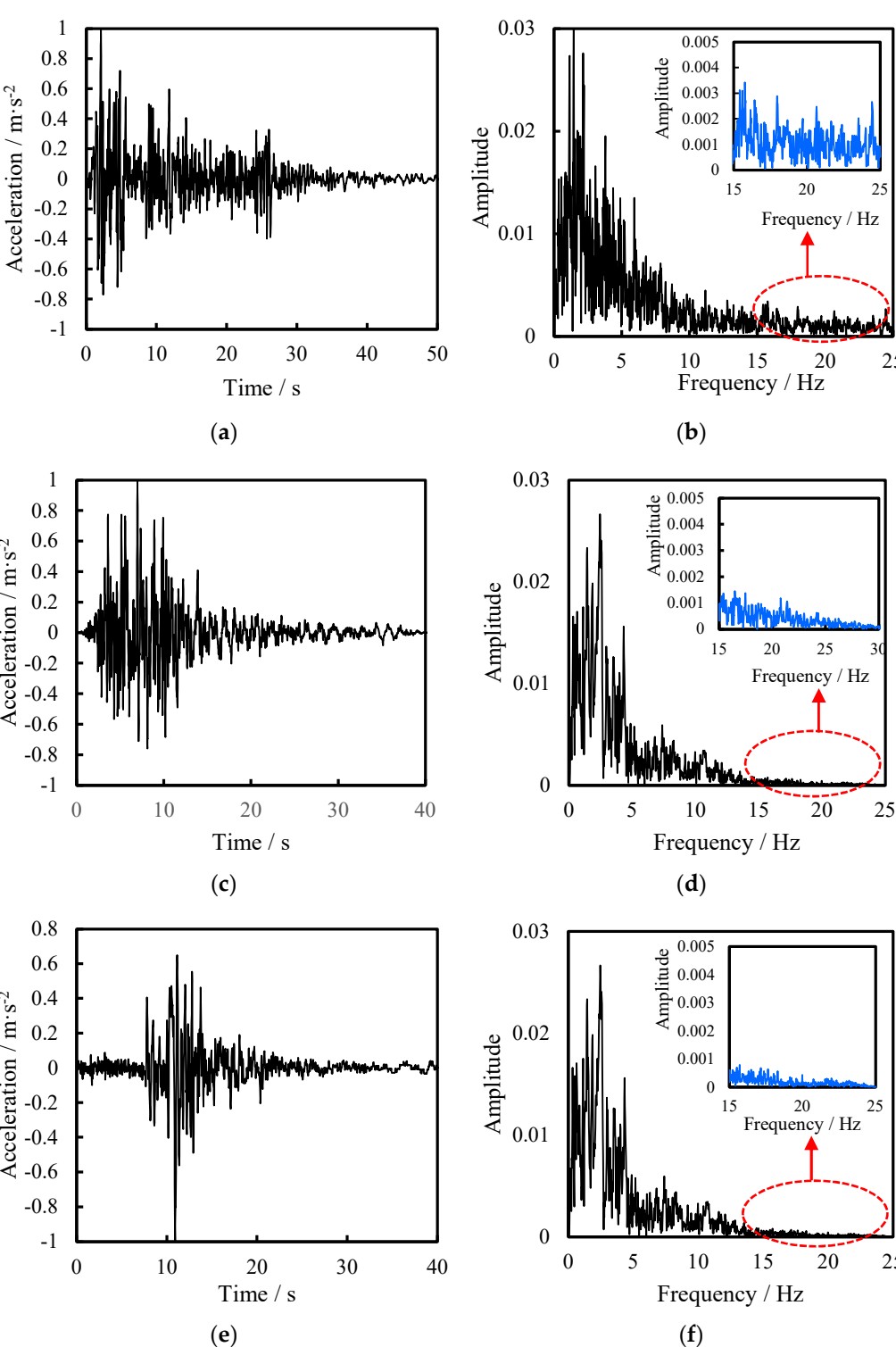

**Figure 24.** Acceleration time-history curves and Fourier spectra. (**a**) El-Centro wave. (**b**) Fourier spectrum under the El-Centro wave. (**c**) Kobe wave. (**d**) Fourier spectrum under the Kobe wave. (**e**) Loma Prieta wave. (**f**) Fourier spectrum under the Loma Prieta wave.

In order to measure the influence of different water depths on the seismic response of the container, the dimensionless coefficient is used to represent the influence coefficient of the dynamic response of the container caused by water depth, and the coefficient of displacement along the sidewall height is as follows:

$$R_d = \frac{d_i - d_0}{d_0} \tag{13}$$

where $d_i$ is the maximum displacement of the sidewall at different water depths, and $d_0$ is the maximum displacement of the sidewall without water. The coefficient of acceleration along the sidewall height is

$$R_a = \frac{a_i - a_0}{a_0} \tag{14}$$

where $a_i$ is the maximum acceleration of the sidewall at different water depths, and $a_0$ is the maximum acceleration of the sidewall without water. The coefficient of von Mises stress along the sidewall height is

$$R_\sigma = \frac{\sigma_i - \sigma_0}{\sigma_0} \tag{15}$$

where $\sigma_i$ is the maximum von Mises stress of the sidewall at different water depths, and $\sigma_0$ is the maximum von Mises stress of the sidewall without water.

The distribution of the maximum displacement and Rd on the sidewall of containers with different liquid levels along the height are displayed in Figures 25 and 26. As the liquid level rises, the maximum displacement and Rd on the sidewall of cylindrical liquid-filled containers increase. When the liquid levels are 0 (no liquid contained), 1, and 2 m, the maximum displacement on the sidewall appears on the top of the container; under conditions with liquid levels of 3 and 4 m, the maximum displacement on the sidewall appears at the height of 2.5 m; if the liquid levels are 5 and 6 m, the maximum displacement on the sidewall occurs at the height of 3 m. Unlike the distribution of the maximum displacement on the sidewall, the position of the maximum Rd remains largely unchanged.

The maximum displacement and Rd on the sidewall have a similar distribution along the height at different liquid levels under the three input seismic waves, and they always increase on the sidewall of the cylindrical liquid-filled container with rising liquid levels, whereas the values of the maximum displacement and Rd are different. Taking the liquid level of 6 m as an example, the maximum displacements on the sidewall are separately 0.052, 0.049, and 0.046 mm under the El-Centro, Kobe, and Loma Prieta waves, and they all appear at a height of 3 m on the sidewall. The Rd values on the sidewall are, separately, 4.90, 4.95, and 4.62 under the El-Centro, Kobe, and Loma Prieta waves, and they all appear at a height of 1 m on the sidewall.

The distribution of the maximum acceleration and Ra on the sidewall of containers with different liquid levels along the height are similar to that of the maximum displacement on the sidewall. Figures 27 and 28 show that as the liquid level rises, the maximum acceleration and Ra on the sidewall of cylindrical liquid-filled containers constantly grow. In the case that the liquid levels are 0, 1, and 2 m, the maximum acceleration on the sidewall appears on the top of the liquid-filled container; if the liquid levels are 3 and 4 m, the maximum acceleration on the sidewall is found at the height of 2.5 m; when the liquid levels are 5 and 6 m, the maximum acceleration on the sidewall occurs at a height of 3 m. The position of the maximum Rd remains almost unchanged.

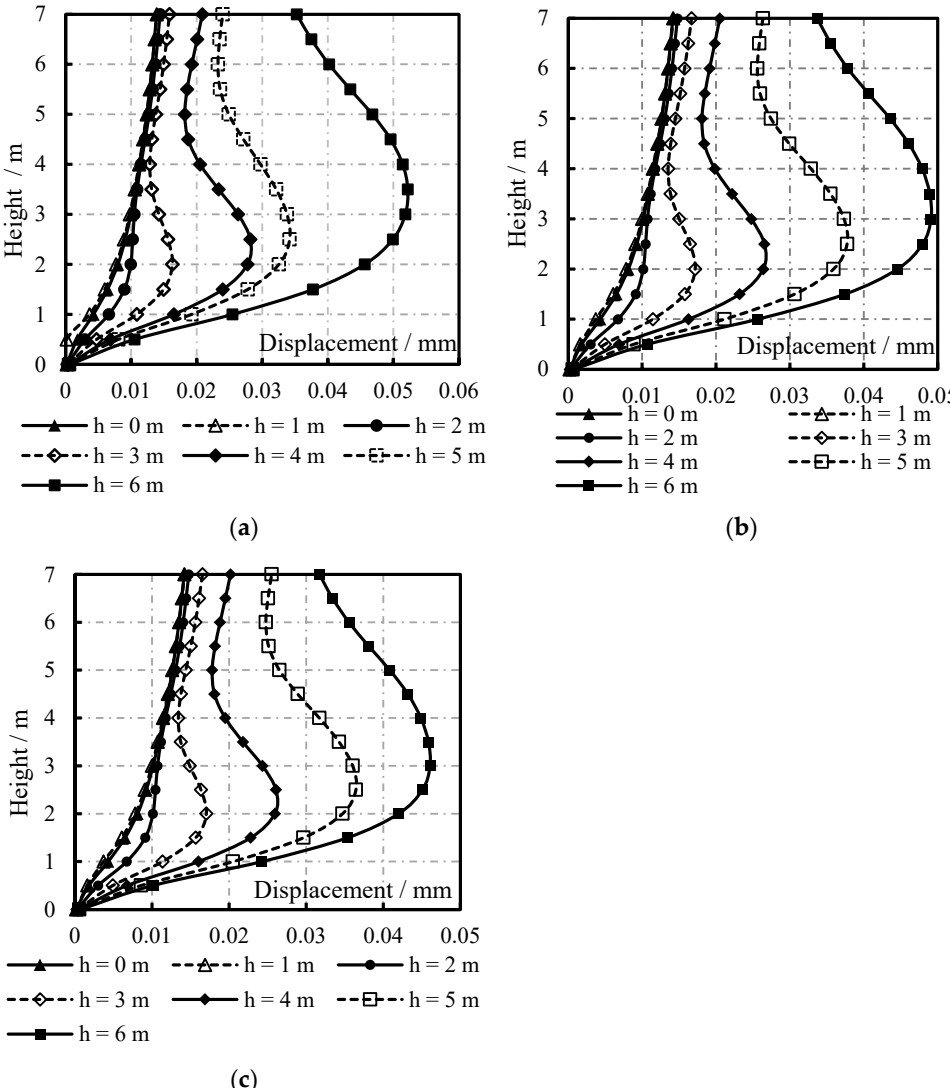

**Figure 25.** The distribution of the maximum displacement on the sidewall of containers. (**a**) El-Centro wave. (**b**) Kobe wave. (**c**) Loma Prieta wave.

Under action of the three seismic waves, the maximum acceleration and Ra on the sidewall of containers with different liquid levels have a similar distribution along the height. The maximum acceleration and Ra on the sidewall of cylindrical liquid-filled containers always increase with rising liquid levels. Taking the liquid level of 6 m as an example, the maximum accelerations on the sidewall are 0.46, 0.20, and 0.058 m/s$^2$ under the El-Centro, Kobe, and Loma Prieta waves, respectively, and they all appear at a height of 3 m. The Ra values on the sidewall are, separately, 4.72, 8.58, and 6.35 under the El-Centro, Kobe, and Loma Prieta waves, and they all appear at a height of 1 m on the sidewall.

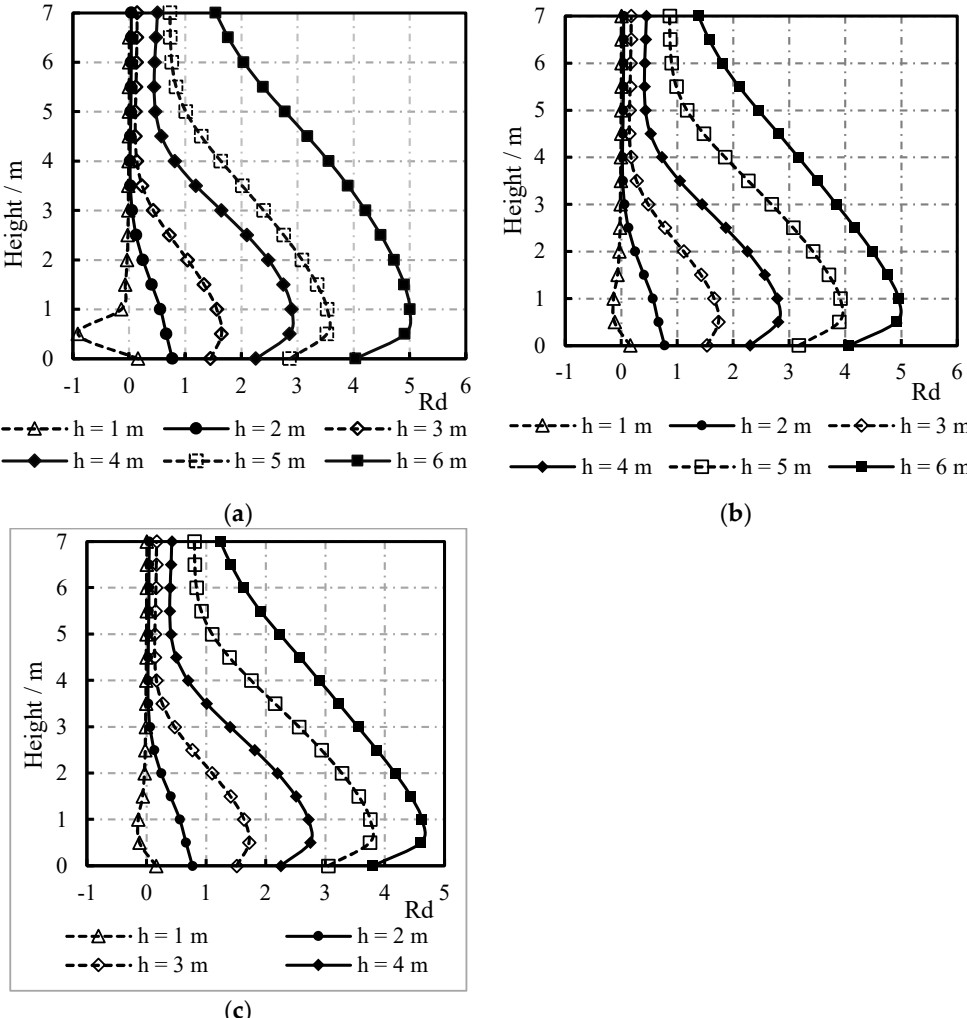

**Figure 26.** The distribution of Rd on the sidewall of containers. (**a**) El-Centro wave. (**b**) Kobe wave. (**c**) Loma Prieta wave.

Under the action of different seismic waves, the maximum acceleration on the sidewall of containers with different liquid levels differs greatly in the distribution height. On the one hand, this is because the frequencies of the input seismic waves differ greatly from the structural frequency of vibration. The main frequencies of the El-Centro, Kobe, and Loma Prieta waves are all concentrated within 5 Hz (Figure 24). In the case of different liquid levels, the minimum and maximum first frequencies of the cylindrical container are 13.17 Hz (liquid level of 6 m) and 19.94 Hz (liquid level of 0 m), respectively. This suggests an unobvious acceleration amplification effect. On the other hand, the El-Centro wave has more high-frequency components (above 15 Hz) compared with the Kobe and Loma Prieta waves, so the maximum acceleration on the sidewall under the El-Centro wave is greater than those under the Kobe and Loma Prieta waves.

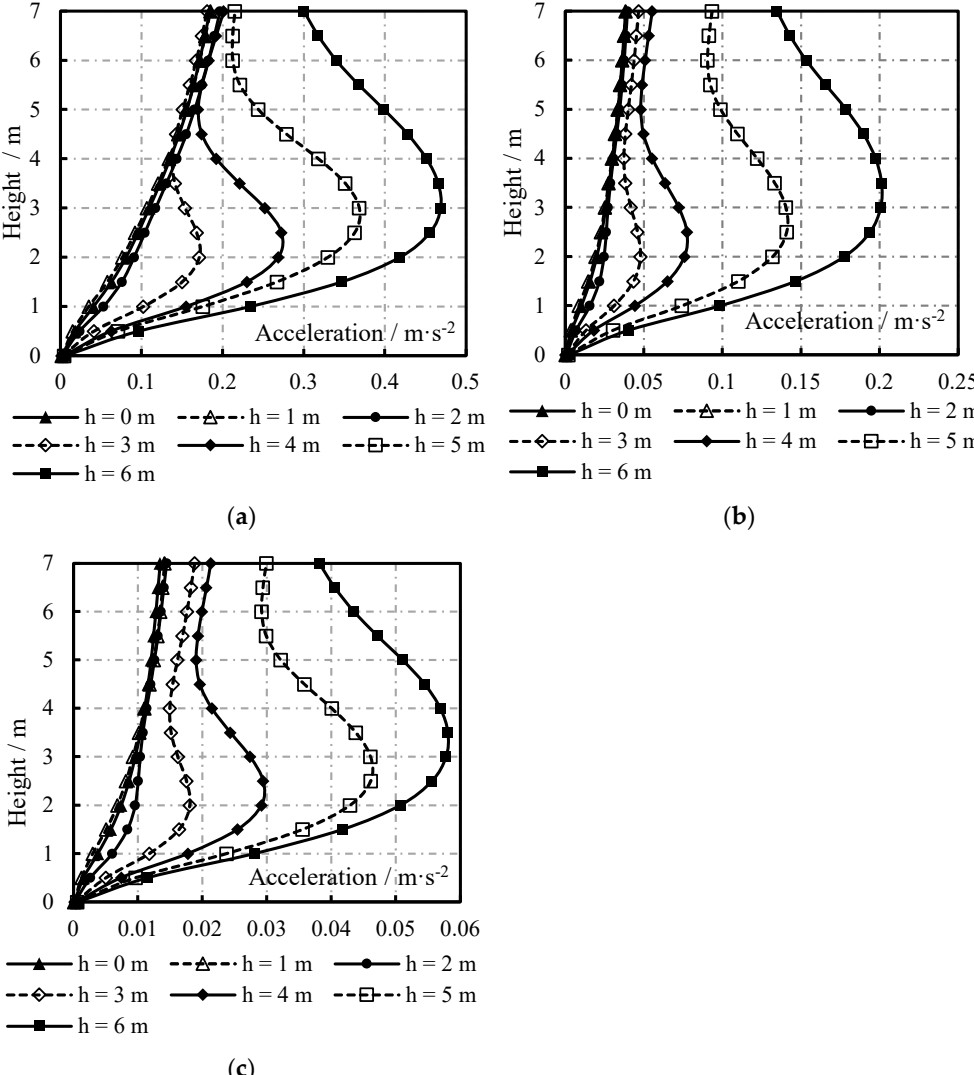

**Figure 27.** Distribution of the maximum acceleration on the sidewall of containers. (**a**) El-Centro wave. (**b**) Kobe wave. (**c**) Loma Prieta wave.

The distribution of the maximum von Mises stress and $R_\sigma$ on the sidewall of containers with different liquid levels along the height are shown in Figures 29 and 30. The maximum von Mises stress is always found at the bottom of the liquid-filled containers. The maximum $R_\sigma$ is always found at approximately 2 m of the liquid-filled containers. Apart from that at the bottom, the maximum von Mises stress on the sidewall of cylindrical liquid-filled containers constantly increases as the liquid rises. When the liquid levels are 0, 1, and 2 m, the maximum von Mises stress on the sidewall appears at a height of 1.0 m; if the liquid levels are 3 and 4 m, the maximum von Mises stress on the sidewall occurs at a height of 2.0 m; in the case that liquid levels are 5 and 6 m, the maximum von Mises stress on the sidewall occurs at a height of 2.5 m.

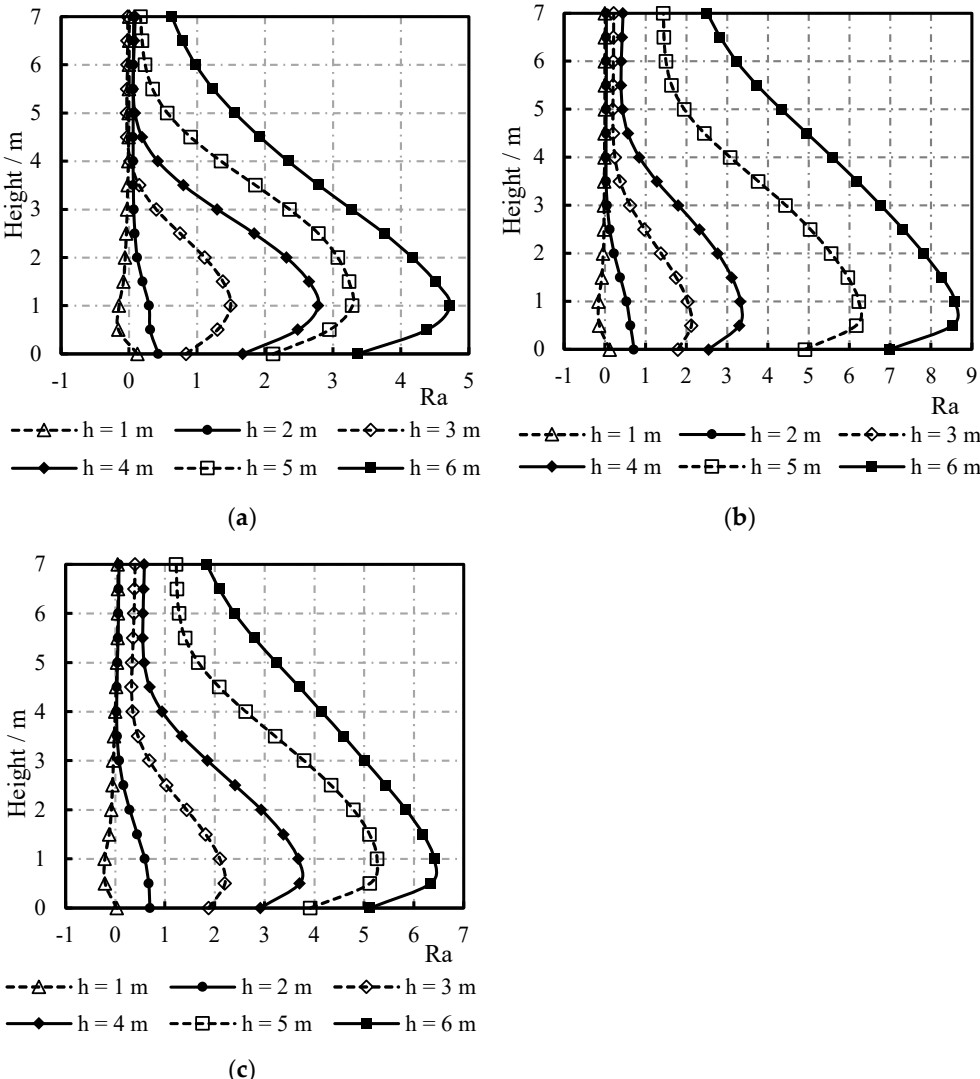

**Figure 28.** The distribution of Ra on the sidewall of containers. (**a**) El-Centro wave. (**b**) Kobe wave. (**c**) Loma Prieta wave.

Under action of the three seismic waves, the maximum von Mises stress and $R_\sigma$ on the sidewall show a consistent distribution along the height of containers filled to different levels. As the liquid level rises, the maximum von Mises stress on the sidewall of cylindrical liquid-filled containers increases, while the values of the maximum von Mises stress and $R_\sigma$ differ. Taking the liquid level of 6 m as an example, under the El-Centro, Kobe, and Loma Prieta waves, the maximum von Mises stresses on the sidewall are, separately, 0.150, 0.144, and 0.135 MPa, and they are all found at a height of 2.5 m. The $R_\sigma$ values on the sidewall are, separately, 6.67, 6.35, and 5.92 under the El-Centro, Kobe, and Loma Prieta waves, and they all appear at a height of 2 m on the sidewall.

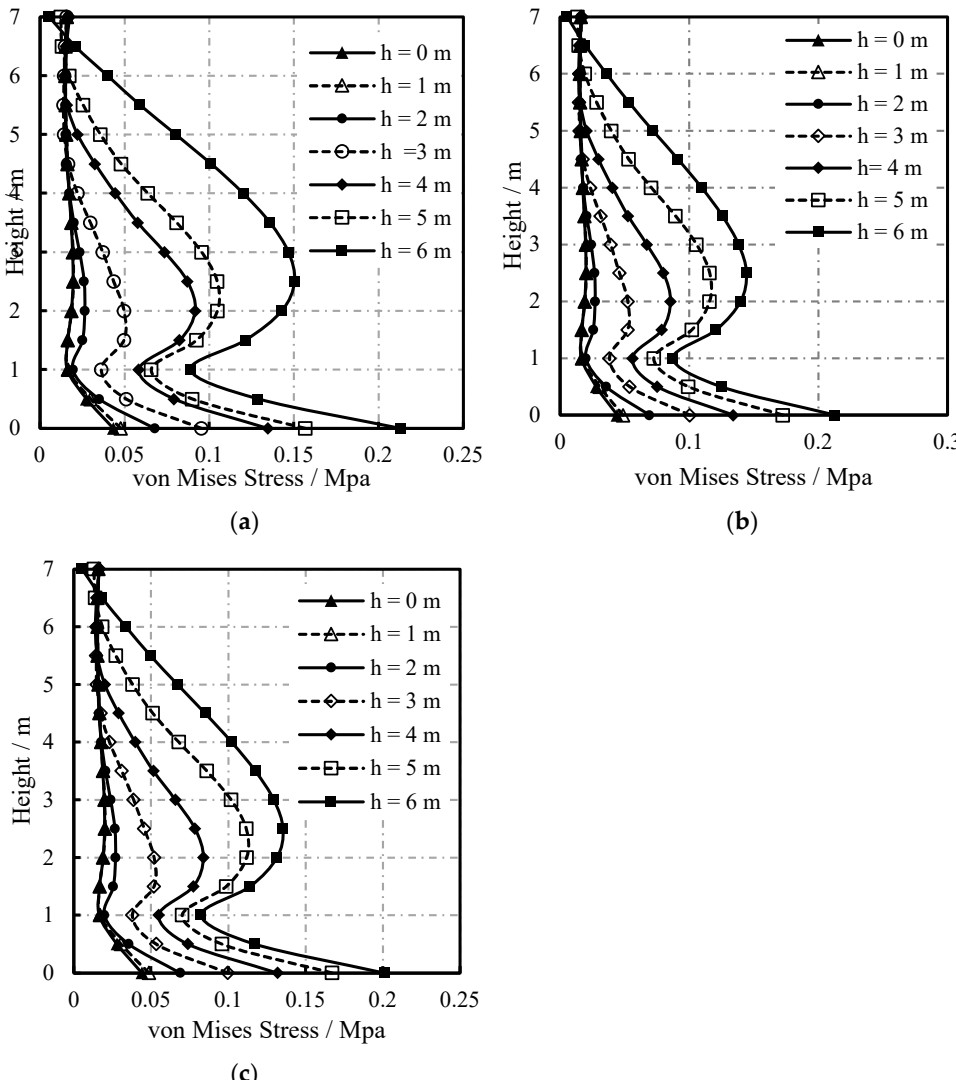

**Figure 29.** Distribution of the maximum von Mises stress on the sidewall of containers. (**a**) El-Centro wave. (**b**) Kobe wave. (**c**) Loma Prieta wave.

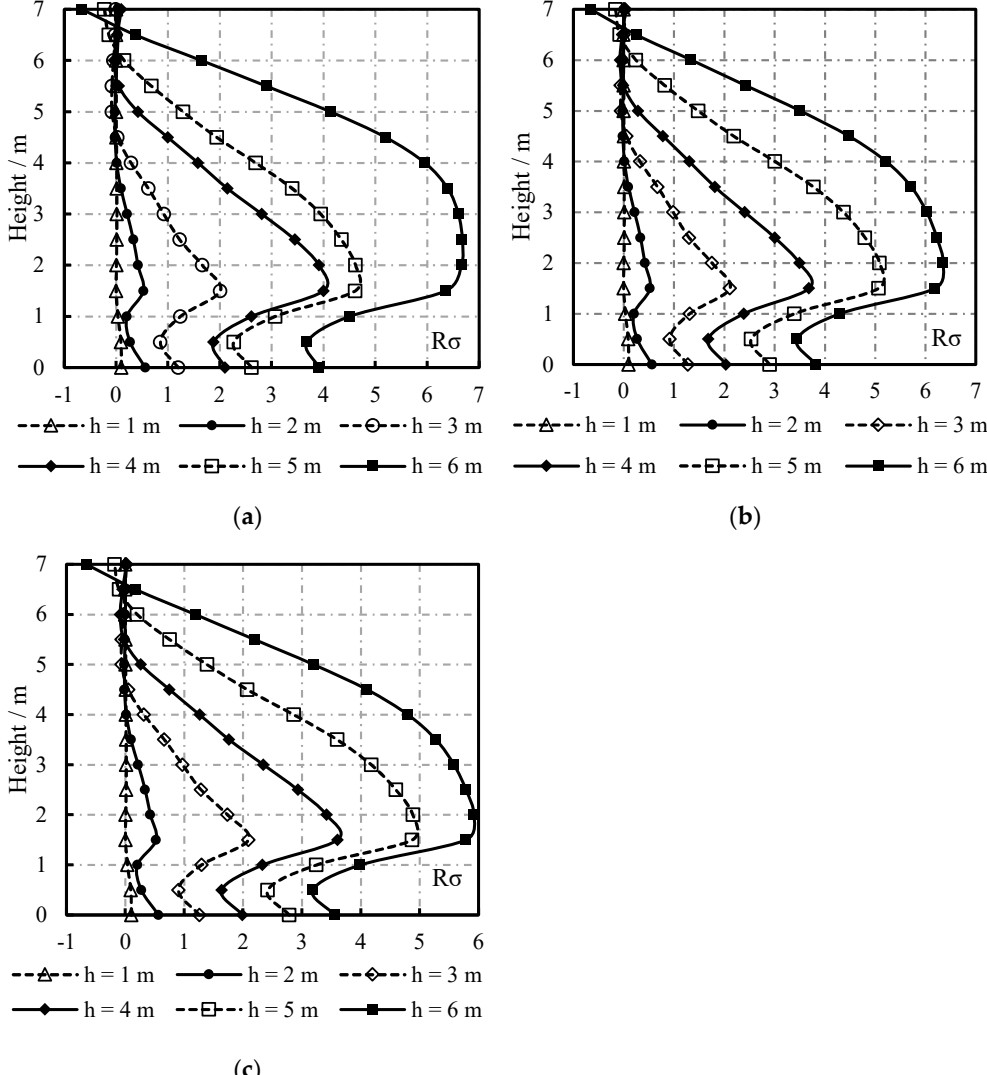

**Figure 30.** The distribution of $R_\sigma$ on the sidewall of containers. (**a**) El-Centro wave. (**b**) Kobe wave. (**c**) Loma Prieta wave.

## 5. Conclusions

A dynamic analysis was performed on liquid sloshing modes in liquid-filled containers and the liquid-filled containers themselves using the FEM based on acoustic fluid elements. The following conclusions can be drawn:

(1) The liquid sloshing modes in 2D and 3D liquid-filled containers of regular shapes and arbitrary cross sections were analyzed and compared with theoretical solutions and test results. The results reveal that the FEM based on acoustic fluid elements is accurate;

(2) The liquid level exerts significant influences on the intrinsic frequency of liquid-filled containers. As the liquid level in liquid-filled containers rises, the vibration frequency of cylindrical containers decreases. When the liquid level in liquid-filled containers is 6 m, the first frequency decreases by 31.24%. During the engineering design of such liquid-filled containers, the influence of the effect of FSIs on the intrinsic frequency of these containers should not be ignored. FEM based on acoustic fluid elements can be used to model such an effect;

(3) For the cylindrical liquid-filled containers in this research, the liquid level essentially did not influence the displacement, acceleration, and stress of the liquid-filled containers under horizontal seismic action if the liquid level was low. As the liquid level rises, the displacement and acceleration of, and stress on, such liquid-filled containers increase

significantly. The acceleration responses of liquid-filled containers are particularly significantly affected by the spectral characteristics of the input seismic wave.

**Author Contributions:** Conceptualization, X.F. and X.B.; methodology, X.F.; software, X.F.; validation, F.Y., Q.Z. and X.B.; formal analysis, X.F.; investigation, Q.Z.; resources, X.F.; data curation, X.B.; writing—original draft preparation, X.F.; writing—review and editing, X.B.; All authors have read and agreed to the published version of the manuscript.

**Funding:** This research was funded by [Hainan Special PhD Scientific Research Foundation of Sanya Yazhou Bay Science and Technology City] grant number [HSPHDSRF-2022-03-008].

**Data Availability Statement:** The raw data supporting the conclusions of this article will be made available by the authors on request.

**Acknowledgments:** The author is very grateful to Wenhui Wei for his careful guidance of this paper.

**Conflicts of Interest:** The authors declare that they have no known competing financial interests or personal relationships that could have appeared to influence the work reported in this paper.

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
