# Peer review of "A Dynamic Analysis Method of Liquid-Filled Containers Considering the Fluid–Structure Interaction"

_applsci, doi:10.3390/app14072688_

Round 1

Reviewer 1 Report

Comments and Suggestions for Authors

Refer to the comments

Comments on the Quality of English Language

It should be supported by the English speaker.

Author Response

January 23, 2024

To whom it may concern.

Thank you for your patience and careful review of the manuscript, we will make comprehensive improvements to the manuscript according to your requirements, and the full revisions are as follows.

Based on the requirements of your esteemed journal, the language usage has been revised by a native English-speaker engaged through the auspices of a professional proofreading service; Both revised version of Language revision using MS-Word “track changes”, and the Language Revision Certificate issued by the service, are all uploaded for your reference.

In addition, the text of the manuscript is marked with purple font to facilitate your review.

The revised image is marked with the name in purple font to indicate that we have made improvements.

The reference format has been refined, and we have not shown the details of the changes.

This work considers the Fluid-Structure interaction problems using an acoustic fluid elements in ANSYS program. Fist of all, manuscript should be supported by the native English speaker. In addition, organization should be checked for more systematic presentation.

Reply:

- Thank you for the suggestion, the article has been comprehensively improved in terms of language.

- Based on the acoustic fluid elements, the dynamic analysis of the liquid sloshing mode and the liquid storage vessel considering the fluid-structure interaction effect is carried out, and the liquid sloshing problem and the influence of the liquid on the liquid storage container itself are comprehensively analyzed.

Firstly, the two-dimensional liquid storage container of the existing test was analyzed to verify the correctness of the method, and then the three-dimensional container was analyzed. This is because the two-dimensional liquid storage container test is easier to implement and more convenient to solve theoretically, and the article starts with the two-dimensional container for analysis, trying to comprehensively analyze the liquid shaking and the influence of the liquid on the liquid storage container itself. The main framework of the article is developed according to this.

Brief comments :

-Original governing equations should be written instead of Eq. 1.  in the ANSYS manual.

Reply:

-Equations (1)-(3) have been rewritten, and the meaning of each symbol in the formula has been explained.

-References in Chines' version should be replaced by the English version because this is an International Journal.

*Reply:

-References have been completely revised according to the format of the journal's specifications, including Chinese literature.

- Captions of Figures and Tables should be written more briefly.

Reply:

Fig. 4 ~ Fig. 9, Fig. 11 ~ Fig. 16, Fig. 11 ~ Fig. 19, each of which has four diagrams, each with its own meaning, is necessary for illustration. Other Captions of Figures and Tables are appropriate.

- Check : Line 109

*Reply:

Thank you for the reminder, the formula number is wrong and has been modified.

- Does the Sec 3.1 include any originality?

Reply:

Section 3.1 is a theoretical formula for the two-dimensional liquid sloshing mode, which is compiled by the author after consulting many works and papers, and is not original to the author, and is cited by the author in the text.

- Typo errors - Line 136: Rize method, Fig10(b),Fig20(c).. Check all part.

*Reply:

The method in the literature is the Rize method, so this article continues to use this statement.

Fig10(b) is a schematic diagram of a spherical container.

Fig20(c). has been modified.

Other figures were also checked.

- What is the meaning of 'Theoretical Calculation' in Table 1 ?

Reply:

The theoretical solutions in Table (1) are the solutions that are solved according to the theoretical formula in Section 3.1. The rectangular container can be solved according to Eq. (4), and other cross-section containers can be solved according to Eq. (6)~(10).

- Figs 4~19 present just a data without any physical meaning.

Reply:

Fig. 4, Fig. 6 and Fig. 8 are the first four sloshing modes of the liquid in different cross-section vessels analyzed by the acoustic fluid element.

Fig. 5, Fig. 7 and Fig. 7 are the first 4th order sloshing modes of the liquid in different cross-section containers measured by the test method.

Fig. 11~16 analyzes the three-dimensional container on the basis of the liquid sloshing mode of the two-dimensional container.

Fig. 18 shows the liquid sloshing mode of a cylindrical liquid storage container.

Fig. 19 shows the mode of the cylindrical liquid reservoir itself.

The actual physical significance of these images is explained in detail in the text.

- Table 3 is just a specific model using the numerical data in Chines language. To be a paper, authors have to obtain the non-dimensionalized data for more generality in Table4.~Figs24.

Reply:

-Table 3 shows the geometric and physical parameters of cylindrical reservoirs. Table 4 shows the liquid shaking frequency of the cylindrical liquid storage container, and Fig. 21~23 show the maximum displacement, acceleration and stress distribution of the side wall under different water depths. The authors argue that even if normalized data are used, the patterns obtained are consistent.

- What is the conclusion for Table 4?

Reply:

Table 4 concludes that the effect of the stiffness of the structure on the liquid sloshing model is negligible, as has been explained in the text.

- This work deals with too many cases for parametric studies only.

Reply:

- Thank you for the suggestion. Based on the acoustic fluid elements, the dynamic analysis of the liquid sloshing mode and the liquid storage vessel considering the fluid-structure interaction effect is carried out, and the liquid sloshing problem and the influence of the liquid on the liquid storage container itself are comprehensively analyzed. Firstly, the two-dimensional liquid storage container of the existing test was analyzed to verify the correctness of the method, and then the three-dimensional container was analyzed. This is because the two-dimensional liquid storage container test is easier to implement and more convenient to solve theoretically, and the article starts with the two-dimensional container for analysis, trying to comprehensively analyze the liquid shaking and the influence of the liquid on the liquid storage container itself. The article is not just some parameterized research.

Reviewer 2 Report

Comments and Suggestions for Authors

Generally

This study conducted dynamic analyses on modes and liquid-filled containers considering the effects of fluid-structure interaction (FSI). Liquid sloshing modes in two-dimensional (2D) and three-dimensional (3D) containers were analyzed, and the results were compared with the liquid sloshing modes measured in experiments and the modes calculated theoretically. The following points need to be examined.

(1)Many studies have examined the swaying and vibration of water in containers using seismic motion. However, the positioning and novelty of this paper in that context were not very clear.

(2) Since a general-purpose software is used, it seems that there is a lack of engineering mechanisms and analysis based on these results. I believe that other general-purpose software could also reproduce the vibration characteristics demonstrated by the numerical simulations, so it is necessary to explain the differences.

1 In the introduction, what is the focus of the examination of the sloshing in the container? If the issue is one of uniform vibration characteristics, please explain the phenomenon that is targeted in the real-world scenario."

2 L114 Is it necessary in this paper to examine using a two-dimensional (2D) model? I think it might be better to start directly with a three-dimensional (3D) approach. Therefore, please explain the necessity. Also, describe the boundary conditions for the liquid and container in the 2D model. What is the reason for choosing square and circular shapes for the containers?

3 L158 It appears that the reproducibility with experiments is quantitative. Please tell me to what extent it has been able to reproduce the results

4  L204

How are the contact conditions between the container and the water handled? It seems that they would significantly vary depending on the boundary conditions and material properties.

5 Fig11-14

Looking at the results, it seems like the fluid is quite viscous. It appears to be a result showing the characteristics of the shape functions of Finite Element Analysis (FEA). Isn't it still limited in terms of representing real-world phenomena?

6 L336 In this analysis, why the liquid level has such an effect on the natural frequency is important. Please examine the mechanism that has been clarified.

7 This study is limited to reproducing specific experimental results, so it is necessary to explain the application limits and the mechanisms that have been identified.

Comments on the Quality of English Language

I can read easily the paper. I think that the paper is not needed to modify English grammar.

Author Response

January 23, 2024

To whom it may concern.

Thank you for your patience and careful review of the manuscript, we will make comprehensive improvements to the manuscript according to your requirements, and the full revisions are as follows.

Based on the requirements of your esteemed journal, the language usage has been revised by a native English-speaker engaged through the auspices of a professional proofreading service; Both revised version of Language revision using MS-Word “track changes”, and the Language Revision Certificate issued by the service, are all uploaded for your reference.

In addition, the text of the manuscript is marked with purple font to facilitate your review.

The revised image is marked with the name in purple font to indicate that we have made improvements.

The reference format has been refined, and we have not shown the details of the changes.

(1)Many studies have examined the swaying and vibration of water in containers using seismic motion. However, the positioning and novelty of this paper in that context were not very clear.

Reply:

- Thank you for the suggestion. Based on the acoustic fluid elements, the dynamic analysis of the liquid sloshing mode and the liquid storage vessel considering the fluid-structure interaction effect is carried out, and the liquid sloshing problem and the influence of the liquid on the liquid storage container itself are comprehensively analyzed.

(2) Since a general-purpose software is used, it seems that there is a lack of engineering mechanisms and analysis based on these results. I believe that other general-purpose software could also reproduce the vibration characteristics demonstrated by the numerical simulations, so it is necessary to explain the differences.

Reply:

In this paper, the two-dimensional liquid storage container of the existing test was analyzed to verify the correctness of the method, and then the three-dimensional container was analyzed. Through the combination of theoretical analysis, experimental analysis and numerical analysis, the mechanical mechanism can be better explained.

The analysis in this paper is based on the acoustic fluid element for the dynamic analysis of the liquid sloshing mode and the reservoir considering the fluid-structure interaction effect. The difference between our analysis and other analyses is that the liquid sloshing and the effect of fluid-structure interaction on the reservoir are analyzed in an acoustic fluid element.

1 In the introduction, what is the focus of the examination of the sloshing in the container? If the issue is one of uniform vibration characteristics, please explain the phenomenon that is targeted in the real-world scenario."

Reply:

The liquid sloshing problem described in the introduction focuses on the use of different methods to solve the sloshing modes of liquids.

1) The aerospace field, such as spacecraft, including rockets, needs to carry large quantities of liquid fuel, and the mass of liquid fuel may be close to half or more of that of spacecraft. Under the excitation of spacecraft motion, the liquid may shake violently, which has an important impact on the motion of the spacecraft.

2) In ship engineering: ships transporting liquids in large ships, such as LNG (Liquefied Natural Gas) ships;

3) Construction engineering: the seismic design of water tower structure and overhead water tank structure involves the sloshing analysis of liquid;

4) Nuclear power engineering: the cooling water tank installed at the top of the containment vessel stores a large amount of cooling water, and the water shaking caused by strong earthquakes will affect the safe operation of the nuclear reactor;

5) Chemical liquid transport vehicles are widely used in loading and unloading acid, alkali, and other corrosive and dangerous liquid media, liquid transport vehicles will cause liquid shaking during the process, and the interaction between vehicles and liquids may pose a threat to driving safety.

And so on, all of which involve the analysis of liquid sloshing and the influence of the liquid itself on the safety of the structure.

2 L114 Is it necessary in this paper to examine using a two-dimensional (2D) model? I think it might be better to start directly with a three-dimensional (3D) approach. Therefore, please explain the necessity. Also, describe the boundary conditions for the liquid and container in the 2D model. What is the reason for choosing square and circular shapes for the containers?

Reply:

Firstly, the two-dimensional liquid storage container of the existing test was analyzed to verify the correctness of the method, and then the three-dimensional container was analyzed. This is because the test of the two-dimensional liquid storage container is easier to implement and more convenient to solve theoretically, and the article starts from the two-dimensional container to analyze, from easy to difficult, trying to comprehensively analyze the liquid shaking and the influence of the liquid on the liquid storage container itself. Therefore, the author believes that two-dimensional analysis is of great significance.

The boundary condition for two-dimensional liquids is a rigid wall boundary, and the same is true for three-dimensional liquids;

The reason for choosing square and circular shapes is that rectangles and reasons are more widely used in engineering.

3 L158 It appears that the reproducibility with experiments is quantitative. Please tell me to what extent it has been able to reproduce the results

Reply:

Experiments can indeed be repeated quantitatively, and the detailed steps of these experiments can be referred to Ref. 22~24 if you are interested.

4  L204 How are the contact conditions between the container and the water handled? It seems that they would significantly vary depending on the boundary conditions and material properties.

Reply:

Thank you for your inquiry, the liquid boundaries in the article are all rigid wall boundaries, and the influence of the liquid boundaries on the sloshing mode of the liquid is not decisive here, because the bulk modulus of the liquid is much smaller than the structure itself.

5 Fig11-14 Looking at the results, it seems like the fluid is quite viscous. It appears to be a result showing the characteristics of the shape functions of Finite Element Analysis (FEA). Isn't it still limited in terms of representing real-world phenomena?

Reply:

In this paper, it is assumed that the fluid is an ideal fluid which is inviscous, irrotational and incompressible, and the influence of the free surface tension of the liquid is not considered.

In addition, it should be noted that for more complex liquid sloshing phenomena, such as breaking waves and splashing, the finite element method based on acoustic fluid elements has its limitations, and it is necessary to use finite volume method, Smoothed Particle Hydrodynamics (SPH) and ALE finite element method.

6 L336 In this analysis, why the liquid level has such an effect on the natural frequency is important. Please examine the mechanism that has been clarified.

Reply:

Due to the fluid-structure interaction, the liquid has a relatively large influence on the reservoir itself. The quality of the liquid in the reservoir is different at different water depths, so the frequency of the reservoir will vary greatly depending on the water depth.

7 This study is limited to reproducing specific experimental results, so it is necessary to explain the application limits and the mechanisms that have been identified.

Reply:

Thank you for your reminding. In this paper, it is assumed that the fluid is an ideal fluid which is inviscous, irrotational and incompressible, and the influence of the free surface tension of the liquid is not considered. For general practical engineering problems, the above assumptions are considered to be feasible. It has been explained in detail in the article.

Reviewer 3 Report

Comments and Suggestions for Authors

This research conducted finite element analysis on liquid sloshing modes at first and compared the results with theoretical measured in tests, thus verifying correctness of the Finite Element Method.

The topic is very interesting, but the paper is missing some fundamental elements.

It is not clear what the correspondence is between the models and the real containers.

The correctness of the solution also depends on how the mesh is constructed. More details would be needed.

Boundary conditions should also be described. The authors talk about FSI analysis. How is it set up? What are the conditions set?

Author Response

February 27, 2024

To whom it may concern.

The revised manuscript is marked in red font to indicate that we have made improvements.

******************************************************************************

Brief comments,

This research conducted finite element analysis on liquid sloshing modes at first and compared the results with theoretical measured in tests, thus verifying correctness of the Finite Element Method.

The topic is very interesting, but the paper is missing some fundamental elements.

It is not clear what the correspondence is between the models and the real containers.                    

Reply:

In the 2D model, Figure 3 (page 6) is the actual model with the geometric parameters given, and Figure 4 (page 7) is the finite element model based on the actual model.

In the 3D model, Figure 11 (page 9) is the actual model, given the geometric parameters, and Figure 12 (page 10) is the finite element model based on the actual model.

Because the stiffness of the container wall is much greater than the stiffness of the liquid, it can be seen from Table 4 (page 15) that it has almost no effect on the sloshing mode of the liquid, so the wall of the liquid sloshing mode is not considered in the analysis of the sloshing mode of the liquid.

The correctness of the solution also depends on how the mesh is constructed. More details would be needed.

Reply:

Page 5 (lines 18~20) and page 9 (lines 15~18) illustrate the mesh size.

It should be noted that after a detailed analysis by the authors, the sloshing frequency and mode of the liquid are not sensitive to the mesh parameters.

Boundary conditions should also be described. The authors talk about FSI analysis. How is it set up? What are the conditions set?

Reply:

Figure 20 (p. 14) has been added to illustrate the interrelationship between containers and liquids.

The boundary conditions for the other models are described in Fig. 4 (p. 7) and Fig. 12 (p. 20).

******************************************************************************

Thank you for your patience and careful review of the manuscript, and the author appreciate it greatly.

Yours.

Sincerely

Round 2

Reviewer 1 Report

Comments and Suggestions for Authors

Refer to comments

Comments on the Quality of English Language

English is not sufficient.

Author Response

January 29, 2024

To whom it may concern.

The revised manuscript is marked in red font to indicate that we have made improvements.

Brief comments,

Organization should be checked for more systematic presentation.

Reply:

It has been reorganized.

-Original governing equations should be written instead of Eq. 1.  in the ANSYS manual.

Reply:

It has been rewritten.

- Captions of Figures and Tables should be written more briefly.

Reply:

Captions of Figures and Tables has been written briefly as possible.

- Typo errors -, Fig10,Fig20. Check all part.

*Reply:

It has been checked.

In addition, 2D and 3D model diagrams are added to make the article more clearly. (Fig 4, Fig 7)

- Table 3 is just a specific model using the numerical data in Chines language. To be a paper, authors have to obtain the non-dimensionalized data for more generality in Table4.~Figs24.

Reply:

-Table 3 has been remade.

The dimensionless coefficient of  has been used.

 - Coefficient of the frequency. (Fig 22)

 - Coefficient of the displacement. (Fig 25)

 - Coefficient of the acceleration. (Fig 27)

 - Coefficient of the von Mises stress. (Fig 29).

******************************************************************************

Thank you for your patience and careful review of the manuscript, and the author appreciate it greatly.

Yours.

Sincerely

Reviewer 2 Report

Comments and Suggestions for Authors

This paper is worthy of publication. Thank you for your support.

Comments on the Quality of English Language

I think that it is enough.

Author Response

January 29, 2024

To whom it may concern.

The revised manuscript is marked in red font to indicate that we have made improvements.

******************************************************************************

Thank you for your patience and careful review of the manuscript, and the author appreciate it greatly.

Yours.

Sincerely

Reviewer 3 Report

Comments and Suggestions for Authors

The paper has improved.

The dimensional correspondence between the real case and the model is not yet clear.

It is not clear what is meant by "fine" mesh, it would be better to provide more details about the mesh.

Author Response

March 2, 2024

To whom it may concern.

The revised manuscript is marked in red font to indicate that we have made improvements.

******************************************************************************

The paper has improved.

The dimensional correspondence between the real case and the model is not yet clear.

Reply:

Thanks to the reviewers' suggestions, we redimensioned the finite element model.

It is not clear what is meant by "fine" mesh, it would be better to provide more details about the mesh.

Reply:

Sorry we didn't use the appropriate word, not "fine mesh", but "refined mesh", which has been modified.

In addition, the boundary conditions of the finite element model are supplemented in section 2.

******************************************************************************

Thank you for your patience and careful review of the manuscript, and the author appreciate it greatly.

Yours.

Sincerely

Round 3

Reviewer 3 Report

Comments and Suggestions for Authors

The paper can be published.